# Does structural social capital lead to proactive green innovation? a three-part serial mediation model

Xinxiang Gao [ID]*

Graduate School of Business, Research and Innovation Management Centre (RIMC), Segi University, Petaling Jaya, Malaysia

* sukd2100480@segi4u.my

## Abstract

Enhancing green innovation for business sustainability represents a pressing global challenge. In the context of the manufacturing industry, the relationship between proactive green innovation (PGI) and structural social capital (SSC) remains a profoundly under-researched area. Drawing upon the theories of social capital and dynamic capability (DC), this study investigated the relationship between SSC and PGI within manufacturing enterprises via three individual and sequential mediating factors, namely cognitive social capital (CSC), relational social capital (RSC), and DC. Adopting a cross-sectional quantitative design, this study collected survey data from 485 manufacturing sector employees in China using purposive sampling. Structural equation modeling analysis of the data revealed no significant direct impact of SSC on PGI, but a strong indirect impact through the sequential mediating influences of CSC, RSC, and DC. The findings suggests that PGI within manufacturing enterprises is not wholly shaped by SSC; rather, firm-level dynamic capabilities, characterized by a sequential mechanism, plays a crucial role in achieving PGI within these enterprises. This paper offers both theoretical and practical contributions and provides recommendations for future research based on its limitations.

## 1 Introduction

The desire to integrate business practices with sustainability arises from the urgent need to address global environmental degradation [1–5]. To this end, green innovation is widely recognized as a potent approach that can yield dual benefits by fostering innovation and spurring positive environmental impacts [6–8]. In theory, "green innovation" refers to the progress achieved through innovations in products or processes aligning with ecological conservation or environmental protection goals [9–11]. Green innovation's simultaneous pursuit of environmental preservation and the creation of commercial value is often seen as an effective strategy for capitalizing on new market opportunities and gaining a competitive edge [12–14]. When organizations actively adopt environment-related innovations to seize such opportunities and competitive advantages in the market, it is known as proactive green innovation (PGI)

**Data Availability Statement:** All relevant data are within the paper and its supporting information files.

**Funding:** The author(s) received no specific funding for this work.

**Competing interests:** The authors have declared that no competing interests exist.

[15,16]. Therefore, exploring the processes influencing the development of corporate PGI carries both theoretical and practical significance.

Recognizing its pivotal importance, researchers have endeavored to identify the relevant antecedents of corporate green innovation from various perspectives [17,18]. For instance, institutional theory underscores the importance of push factors stemming from formal institutions (e.g., regulations, government support) and informal institutions (e.g., social legitimacy), as evidenced in studies by Dong, Wu [19] and Ding [20]. Stakeholder theory, on the other hand, emphasizes the expectations of diverse stakeholders, including customers, non-governmental organizations, and communities [21]. In contrast, the resource-based view highlights the significance of corporate capabilities and resources [22–24]. Sustainability disclosure in emerging markets benefits the development of the green manufacturing industry by Absar, Dhar [25]. While these diverse perspectives shed new light on the drivers of green innovation, it is essential to recognize that corporate green innovation, particularly when it is "proactive," involves the initiative and consciousness of firms. In the context of planned behavior, enterprises' proactive green innovative behavior may not solely be a natural response to external pressures, but rather a deliberate choice. Therefore, social capital and dynamic capability (DC) play indispensable roles in shaping PGI.

Given the complexity and diversity of the social environment, social capital manifests in various dimensions. Consequently, it is imperative to systematically categorize and classify the types of social capital. However, there are gaps in understanding the specific impact of social capital on dynamic capabilities and PGI, particularly when social capital varies across dimensions. Notably, in the existing literature [7,17], there is just one empirical example illustrating a direct relationship between SSC and PGI in the context of manufacturing firms. This connection remains undefined and necessitates further examination in the green innovation literature. Moreover, limited research explores how distinct types of social capital (i.e., structural, relational, cognitive) affect DC, as well as how DC serves as a mediator of the relationship between social capital and PGI [12,26]. In the manufacturing sector, the study of cognitive social capital (CSC), RSC, and DC as mediating mechanisms linking structural social capital (SSC) to PGI remains especially fragmented and scarce [17,27,28].

To address these gaps, this study integrates the theories of social capital and DC to construct and test a theoretical framework examining the sequential mediating effects of CSC, relational social capital (RSC), and DC in the relationship between SSC and PGI within the manufacturing industry. Positing that the three types of social capital (structural, cognitive, and relational) serve as critical conduits for information and resource acquisition [29–31] and exert a positive influence on PGI, we specifically aimed to answer the following questions: (1) What impact does SSC have on PGI? (2) How is SSC linked to CSC, RSC, and DC, ultimately driving PGI in manufacturing enterprises? and (3) In the correlation between SSC and PGI, how do CSC, RSC, and DC individually and sequentially mediate these effects?

To the best of our knowledge, this serial mediation study pioneers in addressing the aforementioned research gaps among Chinese manufacturing enterprises. The findings contribute by revealing the interrelationships among various dimensions of social capital [32], dissecting the underlying paths through which they influence DC and PGI, and exploring the mediating role of DC in this complex framework [33,34].

## 2 Literature and hypotheses

### 2.1 Theoretical focus

Green innovation is widely recognized as the primary avenue for achieving sustainable development [11,22]. It exerts a positive influence on the competitive advantage of enterprises [35],

which is evident in the enterprises' sustainable environmental, economic, and social performance [36]. Through green innovation, enterprises actively reduce environmental pollution, enhance resource utilization efficiency, and elevate their environmental performance. Green innovation adds to the cognitive value of products for customers, which not only translates into profits through the sale of green innovative products but also offsets the costs of green innovation, thereby improving economic performance [19,37]. Furthermore, green innovation bolsters an enterprise's social reputation, leading to improved social performance and, consequently, a competitive edge in the market [38]. Ali, Zahoor [39] and Ali, Zahoor [40] that the petroleum industry continuously adapts to future scenarios by adopting innovative technologies while seeking to mitigate adverse impacts on society and addressing risks associated with climate change to promote sustainable development. Additionally, Zhang and Walton [41] argue that enterprises with more resources and capabilities tend to achieve greater green innovation performance. Ultimately, green innovation successfully reveals the transformation of external knowledge sharing and integration into internal capabilities [7,38,42].

Given its benefits, investigating the drivers of green innovation from various perspectives is a prominent area of research [43]. Firstly, from a capability perspective, Huang and Li [44] assert that an enterprise's dynamic capabilities, coordination abilities, and social reciprocity are vital driving forces for green innovation. Arranz, Arroyabe [45] also highlight the significant impact of innovation capabilities on green innovation. Secondly, adopting an institutional theory perspective, external institutional pressures emerge as the primary influencers of green innovation [43]. Chan, Yee [46] and Li, Zhao [47] respectively argue that environmental regulations and stakeholder legal pressures propel green innovation. Stucki, Woerter [48] contend that mandatory policies such as taxation and regulation may dampen enterprise green innovation initiatives, while incentive policies like subsidies and voluntary agreements can stimulate innovation in green products.

Thirdly, considering the lens of social relationship networks, the green innovation of enterprises is strongly influenced by interactions among major and secondary stakeholders Fliaster and Kolloch [49]. Wakeford, Gebreeyesus [50] posit that social interactions between enterprises and external network members serve as catalysts for green innovation. The more proactive an enterprise is within its social relationship network and the higher the frequency of interactions with network members, the more inclined the enterprise is to engage in green innovation and reap performance benefits [22]. Fabrizi, Guarini [51] contend that research networks such as universities and public research centers positively impact green innovation. Furthermore, Zhang, Tai [52] propose that social capital, as a key conduit for the integration and exchange of green knowledge, exerts a positive influence on green innovation.

Bourdieu and Richardson [53] formally introduced the concept of social capital, which encompasses all resources embedded within a social network characterized by behavioral norms and close interpersonal connections. Social capital is instrumental in assisting individuals and organizations in achieving specific predetermined objectives [54]. In essence, social capital theory views social networks as vital conduits for individuals and entities to access information and resources [30,55,56]. It posits that the collective resources offered by network relationships foster mutual trust among network members across various domains, making it a valuable asset for individuals to leverage [57]. Therefore, this study draws upon social capital theory as one of its foundational frameworks to elucidate the impact of external social capital on the realization of PGI goals within enterprises.

Previous research has already explored the influence of external social capital on various fronts. In the domain of social responsibility, Dhar, Sarkar [58] discovered that the quality of social responsibility information disclosure can be positively adjusted to the relationship between the implementation of green accounting and the sustainable development capabilities

of heavily polluting companies. Gao, Li [59] contend that robust social norms and dense social networks, cultivated by strong external social capital, serve to curtail unethical corporate behavior and enhance resource utilization by enterprises. In the field of strategic management, external social capital aids enterprises in acquiring unique abilities and resources that are challenging for competitors to replicate, thereby strengthening overall competitiveness [60]. Lyu, Peng [61] posit that external social capital has an indirect effect on enterprise innovation by shaping knowledge acquisition. From an innovator's perspective, Zhang, Tai [52] suggest that green innovation within enterprises can be influenced by interactions with a wide array of primary and secondary stakeholders. The engagement between enterprises and external network participants fosters an environment conducive to green innovation [38], thereby enhancing overall green innovation efforts within the enterprise [19,20]. For the purpose of this study, we adopted the framework proposed by Nahapiet and Ghoshal [62], which categorizes external social capital into three components: SSC, CSC, and RSC [26].

According to Teece, Pisano [63], organizations must possess the capability to identify opportunities and threats, acquire valuable resources, and adapt both external and internal resources to navigate the complex and ever-changing business environment [64–67]. This concept is referred to as DC, which can be further subdivided into sensing capability, seizing capability, and reconfiguring capability [68]. DC theory has found application in various management research fields [65,69,70]. Liao, Kickul [71] and Su, Qu [72] suggest that enterprise innovation is realized through the effective utilization of resources, facilitated by dynamic capabilities, while Moroni et al. (2022) argue that DC positively influences enterprise innovation. Feng, Zibibula [65] proposed that innovation is positively affected by DC, although this relationship is often subject to negative regulation by environmental uncertainty [73].

Within the literature exploring the relationship between DC and SSC, DC is frequently seen as a facilitator for integrating external knowledge and resources to help enterprises attain their predefined objectives [74–76]. Qiu, Jie [36] assert that DC plays an intermediary role in the connection between green innovation and competitive advantage. In general, DC is regarded as the primary source of an enterprise's competitive advantage. Enterprises that fail to adapt to their environment often lose a significant portion of their competitive edge due to a lack of DC [77,78]. Grounded in the theoretical perspective of DC, the unique ability of organizations to perceive and effectively integrate information and resources from both internal and external sources becomes the cornerstone for achieving organizational innovation objectives and sustainable competitive advantages [79]. Therefore, this study incorporated DC theory as one of its foundational frameworks to elucidate how firm dynamic capabilities drive the realization of PGI within an enterprise.

## 2.2 Hypotheses development

**2.2.1 Direct effect of SSC on PGI.**   Lyu, Peng [61] argued that external social capital, as a pivotal factor for enterprises in acquiring competitive advantages, facilitates the transfer of knowledge and technical resources, thereby enhancing the innovation performance of enterprises. External social capital is ingrained within the external social relationship network. Enterprises endowed with ample external resources find it easier to access essential key information. Additionally, it reduces bargaining costs, information search expenses, and other transaction-related costs with other network members, thereby directly fostering improvements in technological innovation performance and the formation of an enterprise's competitive edge [80].

Structural ties or connections are a fundamental component of social capital, as the structural aspect creates opportunities for social capital transactions [31,81]. As a type of external

social capital, SSC represents the sum total of various social interaction links within the external social relationship network of a business [82,83]. It provides a comprehensive depiction of interactions among actors, emphasizing the non-personalized aspects of social relationship networks. This dimension focuses on the presence of networks, the overall network structure, as well as the strength and quality of network connections [20,84]. Appiah and Obey [30] assert that when enterprises possess strong SSC, it enables them to outsource non-core activities, thereby streamlining their operations and allowing them to concentrate more on their core functions; ultimately, this enables firms to enhance their competitive advantages.

In the field of innovation management, external social capital is often regarded as a crucial factor influencing enterprise innovation [85]. Guo [7] explored green innovation within enterprises through the lens of social capital and contended that social capital plays a pivotal role in integrating and exchanging green knowledge through social interactions. SSC equips businesses with abundant information and resources, embodied in the interactions (social interactions) between businesses and various entities. These interactions can be analyzed based on factors such as proximity, frequency, the quantity of connections, and the quality of those connections. A higher level of SSC makes it easier for businesses to access green knowledge and intellectual support [20,30]. Moreover, SSC enhances firms' PGI efforts by reinforcing green management practices, green strategic objectives, and green research and development (R&D), as evidenced in prior research. Consequently, we posit that:

H1. SSC significantly influences PGI.

**2.2.2 Individual mediating effect of CSC.** CSC serves as a guiding force for a company's PGI, as it is reflected in the presence of shared vision, shared values, shared language, shared culture, and other such factors among different stakeholders within the company [60,62]. Higher levels of CSC indicate that companies and network members are more aligned with green objectives, facilitating the exchange of information and resources related to green innovation [30]. Companies engaging in more frequent and deeper social interactions are more likely to develop congruent viewpoints during collaborative efforts [86]. In this context, communication among companies within the context of sustainable green development aids in the cultivation of shared ideas and visions among organizations [87].

Tiwana [88] posits that strong relationships and repeated interactions in social networks, referred to as SSC, promote the adoption of rules, encoding, common cognition, and values, collectively known as CSC. Organizations engaging in frequent interactions in collaborations are more inclined to share a similar organizational culture and common value systems compared to those with fewer interactions. As the number of social interactions increases, trust among collaborating companies also develops [89]. Therefore, CSC serves as a bridge between SSC and a company's PGI. Consequently, we hypothesize that:

H2. CSC mediates the effect of SSC on PGI.

**2.2.3 Individual mediating effect of RSC.** Daniel, Midha [90] argued that external social capital does not directly impact the innovation performance of cluster enterprises but rather operates through learning effects and knowledge acquisition. In terms of the relationship between external social capital and PGI, researchers have predominantly explored the factors that encourage green innovation in enterprises from the perspective of external network relationships. However, David, Wu [80] pointed out in their research that the effectiveness of absorbing and applying knowledge from the external environment is contingent upon strong social interactions between enterprises and social entities like users and suppliers. In essence,

the impetus for knowledge spillover emanates from long-term and stable cooperative relationships, thus fostering green initiatives within enterprises.

In this regard, RSC represents the connections between companies and various stakeholders, serving as a catalyst for a company's PGI. It can be assessed in terms of trust, commitment, cooperation, reciprocity, and other facets within the relationships between the company and various actors [60,62]. It underscores the quality of relationships within the external social relationship network of the company. A higher level of RSC correlates with a greater willingness of companies to actively perceive or share information and resources related to green innovation [31,81].

The social interactions fostered by SSC provide avenues for the development and nurturing of RSC. SSC encompasses trust, reciprocity, inter-organizational cooperation, and commitment within the context of green companies, all of which are bolstered by the proximity, frequency, quantity of connections, and intensity of interactions among actors in green businesses [87]. In essence, within the environment and conditions wherein SSC takes shape and configures itself in green enterprises, organizational trust, reciprocity, cooperative relationships, and commitments evolve through the interactions of green business entities over time, thereby facilitating the advancement of PGI in these enterprises.

In this context, the growth of RSC in green enterprises depends on active and repeated social interaction activities among green business entities. In other words, SSC can foster trust between organizations, thereby catalyzing the development of RSC [91]. As such, RSC serves as a mediator between SSC and PGI. Consequently, we hypothesize that:

H3. RSC mediates the effect of SSC on PGI.

**2.2.4 Sequential mediating effects of CSC and RSC.** Social capital theory posits that interactions between organizations foster the development of a shared vision among diverse entities. Shared visions, common values, shared language, shared culture, and the like only emerge in the presence of interactions between businesses [60]. In other words, CSC does not simply materialize spontaneously among different organizations; rather, it emerges when businesses enter specific social relationships and engage in social interactions with other relevant organizations. These interactions transform inter-organizational relationships, guided by common developmental objectives, thereby cultivating trust, reciprocity, and cooperation among green organizations. This creates the necessary resource conditions and settings for green businesses to engage in green innovation. This, in turn, guides the perception, acquisition, and transformation of information, ultimately facilitating the feasibility and effectiveness of firms' PGI efforts [89].

The linkages between green companies and upstream and downstream businesses pave the way for the formation and evolution of CSC between organizations in a unique manner. In essence, when shared visions, shared themes, shared values, and shared culture (i.e., CSC) exist among organizations, they may serve as guiding forces for PGI within those companies. However, if green companies lack sufficient trust in inter-organizational relationships (i.e., RSC) and require dynamic capabilities to perceive, acquire, and transform, it may dampen their enthusiasm for PGI [81].

RSC, as a pivotal factor, minimizes speculative motivations and promotes the actual implementation of PGI within companies [92]. Therefore, SSC can influence enterprise PGI by fostering CSC and, in turn, driving RSC. CSC and RSC can be considered as sequential mediators between SSC and enterprise PGI. Therefore, we hypothesize that within the context of sustainable development, CSC and RSC play sequential mediating roles in the interaction between SSC and PGI. This leads us to propose the following hypothesis:

H4. CSC and RSC sequentially mediate the effect of SSC on PGI.

**2.2.5 Individual mediating effect of DC.**   Wang [93] explored the association between DC and social capital within the framework of social capital theory, contending that social capital can exert a positive influence on DC. In the context of business, DC represents an enterprise's ability to intentionally create, expand, or adapt its pool of resources in response to the evolving demands of the market [73,94]. This DC extends to the context of green innovation, encompassing both product and process innovation, to sustain competitiveness.

The dynamic capabilities of firms are multifaceted, encompassing functions of "sensing," "seizing," and "reconfiguring" to formulate and execute a business model [63]. DC, in the context of our study, refers to a firm's ability to amalgamate resources with the goal of promoting sustainability and environmentally friendly practices within its operations [95]. Teece [96] underscores the importance of a firm's capacity to identify potential opportunities for environmental sustainability and swiftly respond to ecological threats. Firms' ability to adapt rapidly to significant environmental management changes plays a critical role in fostering PGI [97]. Given the increasing recognition of the importance of PGI and environmental sustainability, the enhancement of DC is imperative [98]. Green dynamic firms demonstrate a strong commitment to embracing innovative and sustainable solutions for their clientele, resulting in higher levels of PGI [36]. DC contributes to PGI in various aspects, including enhancements in green product design, environmental management, pollution prevention technologies, waste recycling, and energy conservation [99].

SSC is indicated as a creator of conditions for enterprises to implement dynamic capabilities. It primarily encompasses the closeness of inter-organizational connections, the frequency of connections, the number of connections, and the quality of connections, which altogether positively impact the three facets of DC (i.e., sensing, seizing, and reconfiguring) [100]. Strengthening interactions among such enterprises will motivate DC [83,101]. SSC represents the quantity and closeness of connections among organizations, signifying the channels for their sensing and seizing activities [102]. The more nodes an enterprise has in its social connections (quantity of connections), the more intense its interactions with stakeholders (quality of connections), implying that the enterprise has more scenarios and opportunities for sensing and seizing. Regarding the 'reconfiguring' function, frequent and close interactions between organizations enhance the expectations for cooperative development among different enterprises and dispel misconceptions about enterprise involvement in reconfiguring. Moreover, a greater number of connections can inform enterprises about the quantity and quality of resources they need, reducing the risk of haphazard reconfiguring. High-quality social interaction relationships also increase the likelihood of more participants contributing, sharing, and utilizing resources from other enterprises, which is crucial for innovation and underscores the significance of PGI [75,100].

Taking into account the aforementioned theoretical foundations, we propose that DC explains how SSC affects PGI within the context of sustainable development. Accordingly, we hypothesize that:

H5. DC mediates the effect of SSC on PGI.

**2.2.6 Sequential mediating effects of CSC, RSC, and DC.**   Lyu [85] pointed out that social capital can influence enterprise innovation through dynamic capabilities. Additionally, in their study, Huang and Li [44] contended that DC plays a vital role in driving green innovation, as it helps enterprises navigate environmental changes, allowing them to identify opportunities for green innovation in the market [79]. Notably, the mechanism by which social capital affects enterprise PGI is indirect, relying on the cascading effects of different types of

social capital. SSC, rooted in inter-organizational interactions, directly impacts CSC by stimulating the activities of various actors [20]. It serves as a prerequisite for CSC, which encompasses shared goals, visions, values, culture, and similar factors that facilitate communication and mutual encouragement among various actors, thus establishing the cognitive conditions for the development of inter-organizational trust, cooperation, and reciprocity relationships. Moreover, SSC enhances the exchange of resources and the development of sensing, acquiring, and transforming capabilities between green business partners, thereby improving the effectiveness of DC [77,78].

In addition, RSC influences the link between CSC and DC [79,103], and may serve as a sequential mediator in the effectiveness of DC. Through the reciprocal influence among different dimensions of social capital, companies can cultivate genuine dynamic capabilities, which provide the resource foundation for the development and enhancement of enterprise PGI. Therefore, the cumulative effects of SSC, CSC, and RSC on dynamic capabilities ultimately impact enterprise PGI [27,77].

Considering the discussion above, we hypothesize that CSC, RSC, and DC are sequential mediators of the relationship between SSC and PGI within the context of sustainable development. This leads to the formulation of the following hypothesis:

H6. CSC, RSC and DC sequentially mediate the effect of SSC on PGI.

Fig 1 presents the framework and hypotheses of this study.

## 3 Method

### 3.1 Sampling and data collection

To investigate the proposed relationships among SSC, CSC, RSC, DC, and PGI, purposive sampling was employed to collect survey data from manufacturing small and medium enterprises (SMEs) in China. China, as a large economy, is rapidly working toward achieving the

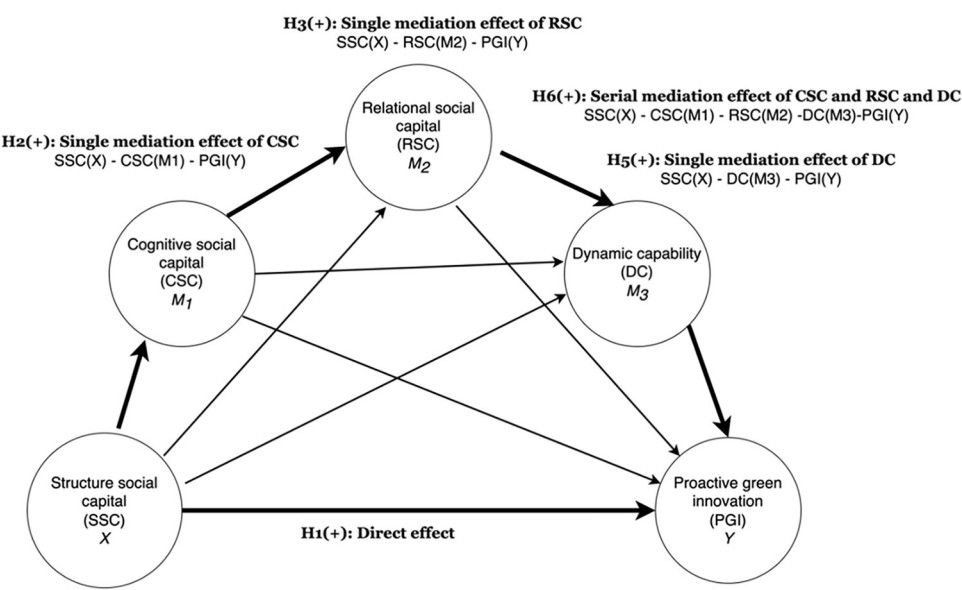

**Fig 1. Research model.**

United Nations' Sustainable Development Goals [104,105]. This has led to the proliferation of SMEs in recent years. However, the COVID-19 pandemic brought about a challenging period for SMEs in China [106]. Many businesses were forced to cease operations permanently, while others incurred significant financial losses. Investigating how Chinese SMEs can thrive in a dynamic business environment, both domestically and internationally, by leveraging Industry 4.0 (I4.0) technology and sustainable business practices has become crucial [107]. In their pursuit of environmental sustainability, these companies are continually implementing technological advancements and engaging in PGI [108,109]. Therefore, it is essential to assess how the PGI of Chinese manufacturing businesses can be enhanced by SSC, CSC, RSC, and DC.

A self-administered questionnaire was developed and pre-tested on 10 manufacturing SME managers and three academics to ensure content validity. Following minor modifications based on the pre-test feedback, the survey was administered to manufacturing SME managers who possessed comprehensive insights into their firms' operations. A total of 620 SMEs in Guangdong Province, Jiangsu Province, Shaanxi Province, and Sichuan Province were sent the survey along with a cover letter outlining the study's objectives and emphasizing voluntary participation. We also assured participants that their responses would remain anonymous and be used solely for academic purposes. Within the data collection period of June 2023 to October 2023, 485 complete and valid surveys were received, yielding a response rate of 78.2%. Male respondents comprised 75.7% of the sample, while female respondents accounted for 24.3%. The majority of participants (87%) had a minimum of one year of experience in their current or most recent R&D managerial role. Moreover, 89% of managers held post-secondary degrees, and the majority (76%) fell within the age range of 25 to 50. Additionally, 51% of the sampled SMEs directly served customers, whereas 24% provided direct services to other businesses and 25% served both customers and businesses.

### 3.2 Measurement instruments

To measure the five research variables (i.e., SSC, CSC, RSC, DC, and PGI), a total of 30 measurement items were adopted from previous research and adapted to suit this study's context. We assessed SSC and CSC using a set of five and four items, respectively, adapted from Ortiz, Donate [86]. RSC was measured using five items taken from Prieto-Pastor, Martín-Pérez [89], whereas DC was evaluated with 12 items from Wilden, Gudergan [110]. Finally, PGI was measured using a set of four items adapted from [15] and Cao, Shen [6]. As a supplement to these items, we added a question asking respondents to score their company's PGI over the past three years, ranging from "much worse" to "much better." Please refer to S1 Table for the questionnaire items. The questionnaire also solicited demographic information on respondents' position, age, level of education, and years of experience, among others. Except for the demographic section, participants rated all the items on a 7-point Likert scale, with a score of one indicating "strongly disagree" and a score of seven indicating "strongly agree".

### 3.3 Common method bias

To mitigate potential common method bias (CMB) during the data collection process, we implemented procedural adjustments following the recommendations of Podsakoff, MacKenzie [111]. First, the survey provided clear instructions on how to answer the questions and urged respondents to provide their most accurate responses while ensuring confidentiality, anonymity, and voluntary participation. We also emphasized that there were no right or wrong answers. Additionally, we carefully reviewed each item to ensure clarity and conciseness, avoiding any unusual or ambiguous language. We also altered the sequence of statements to minimize the likelihood of respondents making educated guesses [112]. In this manner, we

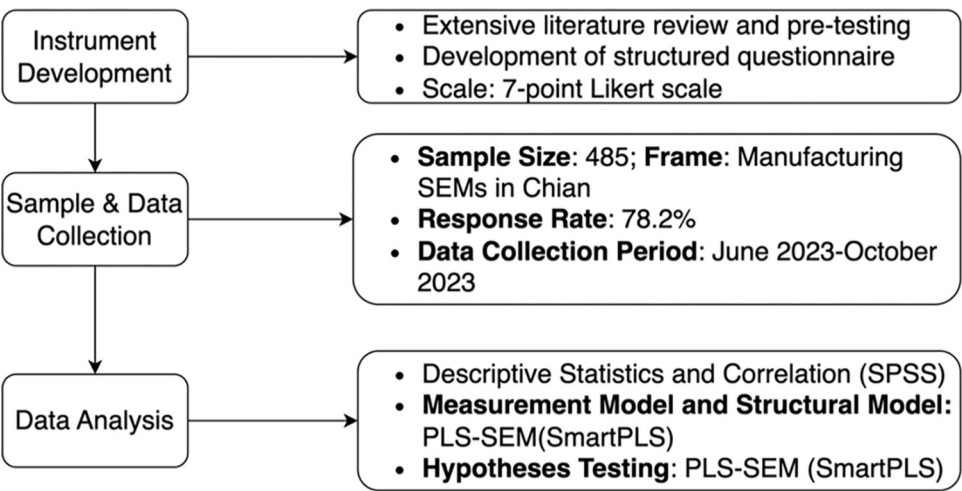

**Fig 2. Flowchart of research materials and methods.**

ensured that our measurement items were clear and comprehensible. In terms of statistical remedies for CMB, we conducted a post-hoc assessment using Harman [113] single-factor test. The analysis indicated that the highest variance explained by a component was only 39.87%, which is less than 50%. This finding confirms the absence of CMB in our data.

## 3.4 Data analysis technique

Partial least squares structural equation modeling (PLS-SEM) was utilized via the SmartPLS software to analyze the cross-sectional data obtained from Chinese manufacturing SMEs. In contrast to basic regression, SEM involves multiple independent variables and predictors (Astrachan et al., 2014), rendering it a more comprehensive approach [114]. Hair, Hult [115] suggest that for SEM to provide precise and reliable estimates, a minimum of 100 input data points is necessary; our dataset comprised 485 records, exceeding this requirement. Furthermore, Gpower analysis is strongly recommended for determining the appropriate sample size in the PLS literature [115]. Therefore, we utilized G*Power 3.1.9.7 to determine the minimum sample size. With 10 predictors, an alpha value of 0.05, an effect size of 0.15, and a power level of 0.80, our sample size of 485 exceeded the minimum of 118 required by G*Power.

Non-response bias analysis was conducted using the methodologies described in earlier studies [116]. Specifically, Pearson's chi-square test for discrete variables was used to compare firm age, firm size, education, and experience between early and late responders [117]. The results indicated no significant differences between individuals who responded early and those who responded late. Therefore, non-response bias was not a concern in this research. Fig 2 illustrates the flow of research materials and methods in this study.

## 4 Results

### 4.1 Descriptive analysis results

Table 1 shows the results of our descriptive and correlation analyses. The descriptive statistics indicated above-average mean scores for RSC, DC, SSC, CSC, and PGI, which were 4.557, 4.723, 4.473, 4.487, and 4.892, respectively. Consistent with prior studies, the skewness and kurtosis parameters were below 3.0 and 10.0, respectively [118]. Additionally, the highest

**Table 1. Descriptive statistics and correlation analysis.**

|  | Mean | SD | Skewness | Kurtosis | 1 | 2 | 3 | 4 | 5 |
|---|---|---|---|---|---|---|---|---|---|
| 1. SSC | 4.473 | 0.693 | 0.639 | -0.398 | 1 |  |  |  |  |
| 2. CSC | 4.487 | 0.714 | 0.418 | -0.743 | 0.541 | 1 |  |  |  |
| 3. RSC | 4.557 | 0.762 | 0.212 | -1.021 | 0.743 | 0.581 | 1 |  |  |
| 4. DC | 4.723 | 0.810 | 0.103 | -1.319 | 0.612 | 0.714 | 0.621 | 1 |  |
| 5. PGI | 4.892 | 0.873 | -0.116 | -1.674 | 0.593 | 0.643 | 0.602 | 0.721 | 1 |

[a] Note: SSC = SSC, CSC = CSC, RSC = RSC, DC = DC, PGI = PGI

[b]Source: Authors' calculation.

correlation among the underlying factors was 0.721, confirming the absence of a high correlation and supporting the suitability of the model for subsequent statistical exploration.

## 4.2 Measurement model results

The first stage in PLS-SEM is the assessment of the measurement model, wherein four tests are designed to verify item-level reliability, internal consistency reliability, convergent validity, and discriminant validity. First, the minimum and maximum factor loadings demonstrated in Table 2 are 0.617 and 0.902, respectively, surpassing the threshold of 0.50 recommended by Hair, Sarstedt [119] and Hair, Hult [115]. This result indicates that this research had adequate item-level reliability. Next, according to Rahman, Azma [120], the internal consistency reliability of each variable should be ascertained using Cronbach's alpha and composite reliability (CR), both of which should exceed the minimum threshold of 0.70 [121]. The Cronbach's alpha and CR values shown in Table 2 surpass this criterion, signifying that all the constructs were consistent, internally cohesive, and reliable.

Third, convergent validity assesses the extent to which different items are expected to be associated with an identical construct. Table 2 shows that the average variance extracted (AVE) values ranged from 0.541 to 0.715, meeting the minimum criterion of 0.50 set by Hair, Risher [122]. Therefore, convergent validity was established in this study. Lastly, discriminant validity ensures that two indicators do not share a statistical identity [122]. Fornell Fornell and Larcker [123] introduced a conventional benchmark to assess discriminant validity, wherein the square root of AVE values is compared against squared correlation values. However, Henseler, R. Sinkovics [124] recommended the heterotrait-monotrait ratio (HTMT) of correlations as an innovative method to assess discriminant validity, asserting that the traditional metric is not suitable. They recommended setting the HTMT critical point to 0.90 for the same concepts in the theory and 0.85 for conceptually different variables.

In this paper, both the HTMT criterion and the standard Fornell-Larcker criterion were utilized. Table 3 shows that the square root of the AVE for each factor exceeded the correlation coefficients in each row, while Table 4 indicates that the HTMT values for all constructs were below 0.85. Both these results confirm the establishment of discriminant validity as per their respective criterion. In addition, Hair, Risher [122] suggested that the value of the Variance Inflation Factor (VIF) to evaluate multicollinearity should not exceed 5.0. The test results indicated that the VIF values were all below 5, meeting the requirement for discriminant validity (see Table 4). Fig 3 depicts the measurement model results.

The predictive validity of the constructs is presented in Table 5, demonstrating the ability of the independent constructs in our model to make predictions of the dependent ones. Two metrics, $R^2$ and $Q^2$, were employed to assess predictive accuracy. According to Cohen [125], a substantial $R^2$ should exceed 0.26. The serial mediation model accounted for 46.8% of the

**Table 2. Construct validity and reliability.**

| Items | Factor Loadings | Alpha | CR | AVE |
|---|---|---|---|---|
| SSC1 | 0.841 | 0.895 | 0.896 | 0.704 |
| SSC2 | 0.805 | | | |
| SSC3 | 0.829 | | | |
| SSC4 | 0.847 | | | |
| SSC5 | 0.872 | | | |
| CSC1 | 0.833 | 0.867 | 0.868 | 0.715 |
| CSC2 | 0.861 | | | |
| CSC3 | 0.880 | | | |
| CSC4 | 0.806 | | | |
| RSC1 | 0.821 | 0.889 | 0.892 | 0.693 |
| RSC2 | 0.789 | | | |
| RSC3 | 0.800 | | | |
| RSC4 | 0.845 | | | |
| RSC5 | 0.902 | | | |
| DC1 | 0.731 | 0.922 | 0.927 | 0.541 |
| DC2 | 0.750 | | | |
| DC3 | 0.777 | | | |
| DC4 | 0.679 | | | |
| DC5 | 0.805 | | | |
| DC6 | 0.639 | | | |
| DC7 | 0.783 | | | |
| DC8 | 0.732 | | | |
| DC9 | 0.756 | | | |
| DC10 | 0.812 | | | |
| DC11 | 0.617 | | | |
| DC12 | 0.722 | | | |
| PGI1 | 0.863 | 0.848 | 0.853 | 0.687 |
| PGI2 | 0.784 | | | |
| PGI3 | 0.806 | | | |
| PGI4 | 0.860 | | | |

[a] Note (s): Alpha = Cronbach's Alpha, CR = Composite reliability, AVE = Average variance extracted.
[b] Source: Authors' calculation.

**Table 3. Fornell–Larcker criterion.**

| | SSC | CSC | RSC | DC | PGI |
|---|---|---|---|---|---|
| SSC | 0.839 | | | | |
| CSC | 0.685 | 0.846 | | | |
| RSC | 0.665 | 0.649 | 0.833 | | |
| DC | 0.624 | 0.635 | 0.669 | 0.880 | |
| PGI | 0.529 | 0.557 | 0.578 | 0.736 | 0.829 |

[a] Note: SSC = SSC, CSC = CSC, RSC = RSC, DC = DC, PGI = PGI; Bold values on the correlation matrix's diagonal are AVE's square roots. Off-diagonal elements below the diagonal are correlations among the constructs. [b] Source: Authors' calculation.

**Table 4. HTMT criterion.**

|       | SSC   | CSC   | RSC   | DC    | VIF |
|-------|-------|-------|-------|-------|-----|
| CSC   | 0.775 |       |       |       | 1   |
| RSC   | 0.744 | 0.736 |       |       | 1   |
| DC    | 0.670 | 0.695 | 0.722 |       | 1   |
| PGI   | 0.606 | 0.651 | 0.662 | 0.822 | 1   |

[a] Note: SSC = SSC, CSC = CSC, RSC = RSC, DC = DC, PGI = PGI. [b] Source: Authors' calculation.

variance in CSC, 51.4% in RSC, 54.3% in DC, and 77.2% in PGI, verifying substantial $R^2$. Furthermore, $Q^2$ indicates the predictive relationship among internal variables, where a value above zero suggests predictive relevance. The results confirmed the predictive relevance of the variables examined in this study, with all $Q^2$ exceeding zero (CSC = 0.466, RSC = 0.440, DC = 0.391, and PGI = 0.277). Moreover, the serial mediation model, involving a sequence of three consecutive mediators (i.e., CSC, RSC, and DC) in the link between SSC and PGI, exhibited satisfactory fit statistics. The model fit was considered acceptable, with the PLS-SEM SRMR coefficient at 0.091, which is below the threshold of 0.10.

## 4.3 Structural model results

After establishing the measurement model, we proceeded to analyze the structural model using the bootstrapping technique in SmartPLS 4.0 with 10,000 subsamples. The inner model, used to assess the proposed hypotheses, calculates both the p-value and t-value. A hypothesis is considered supported if the p-value is below 0.05 or the t-value exceeds 1.96. The results of the analysis and corresponding hypotheses are presented in Table 6 and Fig 4.

**4.3.1 Direct effect and individual indirect effect results.** It can be observed from the analysis that the positive influence of SSC on PGI was not statistically significant (β = -0.039, t = 1.391, and p = 0.164), leading to the rejection of H1. Similarly, the individual mediating

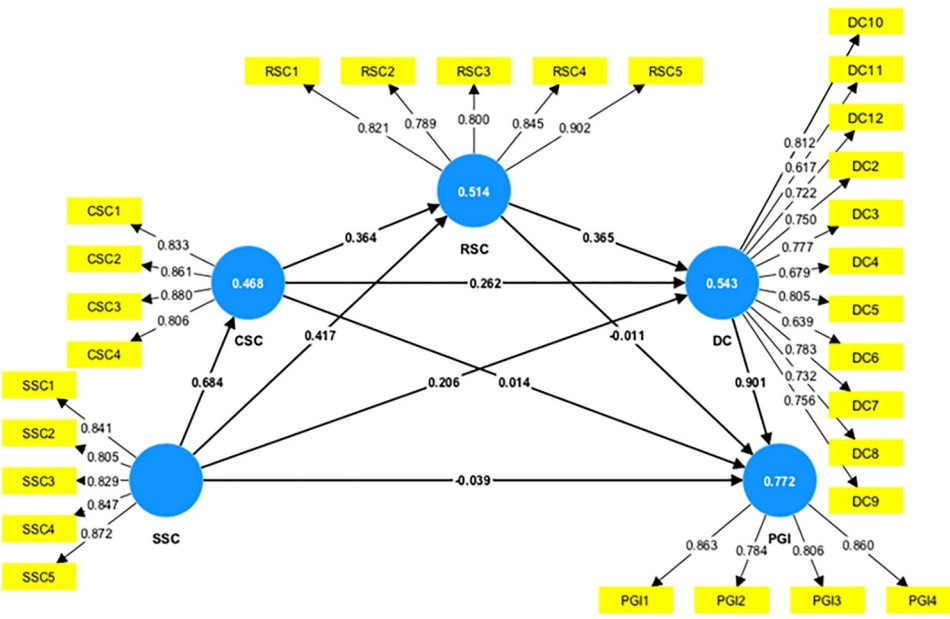

**Fig 3. Measurement model estimation results.**

**Table 5. Predictive relevance of the model.**

|  | R Square | Q2 (= 1-SSE/SSO) |
|---|---|---|
| CSC | 0.468 | 0.466 |
| RSC | 0.514 | 0.440 |
| DC | 0.543 | 0.391 |
| PGI | 0.772 | 0.277 |

[a] Source: Authors' calculation.

effects of CSC (a1b1; $\beta = 0.010$, t = 0.424, and p = 0.672) and RSC (a2b2; $\beta = -0.005$, t = 0.343, and p = 0.732) between SSC and PGI were not statistically significant. Therefore, both H2 and H3 did not receive support. Considering these results, it can be concluded that SSC does not enhance PGI, either directly or indirectly through cognitive and RSC.

However, the indirect impact of SSC on PGI via DC (a3b3; $\beta = -0.185$, t = 4.318, and p<0.001) was found to be statistically significant. Hence, H5 was supported. Given the absence of a significantly positive direct effect, this finding indicates the full mediation of DC in the relationship between SSC and PGI.

**4.3.2 Sequential mediation effect results.** The findings further show that the indirect effect of SSC on PGI through the sequential mediation of CSC and RSC (a1d2b2; $\beta = -0.003$, t = 0.340, and p = 0.734) was non-significant. Therefore, H4 was not supported. Considering the results thus far, it appears that social capital (i.e., SSC, CSC, and RSC), as a whole, has neither a direct nor indirect significant impact on PGI.

Nonetheless, H6 was supported, as the indirect impact of SSC on PGI through the sequential mediation of CSC, RSC, and DC was both significant and positive (a1d21d32b3; $\beta = 0.082$,

**Table 6. Direct and indirect hypotheses testing results.**

| Hypotheses | Structural Path | Coefficient | t-statistics | RIETE (%) | Hypotheses test result |
|---|---|---|---|---|---|
|  | Total effect |  |  |  |  |
|  | SSC(X) → PGI(Y) (c) | 0.529*** | 16.369 |  |  |
|  | Direct effect |  |  |  |  |
| H1 | SSC(X) → PGI (Y) (c') | -0.039[ns] | 1.391 | -7.3 | Not supported |
|  | Indirect effect |  |  |  |  |
| Total indirect effect |  | 0.568*** | 15.558 | 107.3 |  |
|  | Single mediation effect |  |  |  |  |
| H2 | SSC(X) → CSC (M1) → PGI (Y) (a1b1) | 0.010[ns] | 0.424 | 1.8 | Not supported |
| H3 | SSC(X) → RSC (M2) → PGI (Y) (a2b2) | -0.005[ns] | 0.343 | -0.9 | Not supported |
| H5 | SSC (X) → DC (M3) → PGI (Y) (a3b3) | 0.185*** | 4.318 | 35 | Supported |
|  | Sequential mediation effect of two mediators |  |  |  |  |
| H4 | SSC(X)→CSC(M1)→RSC(M2)→PGI(Y) (a1d21b2) | -0.003[ns] | 0.340 | -0.6 | Not supported |
|  | SSC(X)→CSC(M1) → DC (M3) →PGI (Y) (a1d31b3) | 0.162*** | 5.878 | 30.6 |  |
|  | SSC(X)→RSC(M2)→DC(M3)→PGI(Y) (a2d32b3) | 0.137*** | 5.983 | 25.9 |  |
|  | Sequential mediation effect of three mediators |  |  |  |  |
| H6 | SSC(X)→CSC(M1)→RSC(M2)→DC (M3)→PGI(Y)(a1d21d32b3) | 0.082*** | 5.126 | 15.5 | Supported |

[a] Notes: SSC = SSC; CSC = CSC; RSC = RSC; DC = DC; PGI = PGI; ns = not significant.

*p <0.05

**p <0.01

***p <0.001 (two-tailed test). [b] Source: Authors' calculation

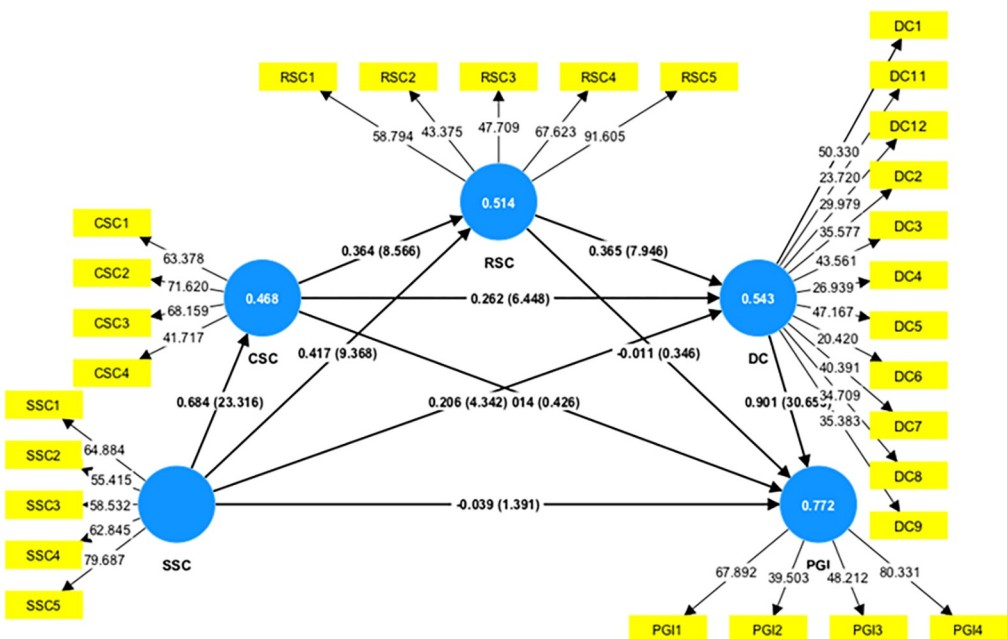

**Fig 4. Structural model estimation results.**

t = 5.126, and p<0.001). This result supports the four-path sequential mediation effect involving CSC, RSC, and DC in the connection between SSC and PGI. In short, CSC, RSC, and DC collectively and sequentially mediate the relationship between SSC and PGI, notwithstanding the insignificant direct effect of SSC on PGI.

**4.3.3 Total effect decomposition between SSC and PGI.** The total effect between SSC and PGI is decomposed in Table 6. The respective total effect, direct effect, and indirect effect in the correlation between SSC and PGI were 0.529, -0.039, and 0.568, respectively. The proportion of indirect effects (107.3%) in the total effect was notably larger than that of the direct effect (7.3%), underscoring the significance of the mediating roles played by CSC, RSC, and DC in the relationship between SSC and PGI.

Within the total indirect effects of 107.3%, the respective proportions attributed to CSC, RSC, and DC were 1.8%, -0.9%, and 35%, respectively. These findings indicate that DC, as a sole individual mediator in the relationship between SSC and PGI, plays a role approximately 20 times more significant than CSC and far more substantial than RSC. In fact, CSC and RSC were insignificant in individually mediating the relationship between SSC and PGI.

The sequential mediators (CSC, RSC, and DC) contributed 15.5% of the total indirect effect of 107.3% in the impact of SSC over PGI. This outcome holds significance, as it validates that CSC, RSC, and DC are the three meaningful consecutive mediators for SSC to influence PGI, even though their share of total indirect effects was relatively modest.

## 5 Discussion

This study sheds light on the direct and mediating relationships between SSC and PGI, an area that has received relatively limited attention in prior research. The results emphasize that CSC, RSC, DC, and SSC collectively play crucial roles in achieving PGI within the manufacturing sector.

Our findings highlight the significant impact of DC as a single mediator in the relationship between SSC and PGI, aligning with the conclusions of previous scholars [20,26,28,126–128].

It is evident that DC is instrumental in implementing a manufacturing firm's policies and strategies, enabling them to adapt to volatile business environments and fostering innovation [22,36,129]. The sensing, seizing, and reconfiguring activities enabled by dynamic capabilities can effectively harness the potential of social capital to enhance PGI in manufacturing enterprises, illustrating that social capital can drive PGI primarily through DC.

Furthermore, the findings indicate that CSC and RSC have an insignificant mediating influence on the relationship between SSC and PGI, whether individually or in sequence; this implies a more nuanced relationship [30]. Indeed, our study reveals that DC is an imperative component in the sequential mediation to collectively drive PGI, as the results supported that CSC, RSC, and DC sequentially influence the relationship between SSC and PGI.

These findings align with the arguments put forth by Alguezaui Alguezaui and Filieri [82] and Jääskeläinen, Korhonen [31] that SSC initiates CSC and RSC, subsequently triggering green reactions like DC within manufacturing enterprises in a sequential mechanism [68,128]. The extent of social interaction and the strength, frequency, quantity, and quality of social connections among organizations reflect SSC practices [59,60]. Consequently, PGI serves as a sequential manifestation of CSC, RSC, DC, and SSC implementation.

## 5.1 Theoretical contributions

This study makes significant contributions to the current manufacturing literature in various aspects. Firstly, it integrates social capital and DC theories, providing a novel theoretical perspective for understanding the relationship between SSC and firm PGI. Notably, this study uncovers an implicit connection between social capital theory and DC theory, exploring how the practices of SSC, as proposed in social capital theory [130], simultaneously enhance manufacturing enterprise capabilities and motivation, as suggested by DC theory [68,128]. Such integration fosters CSC and RSC, facilitating the sharing of green knowledge among manufacturing enterprises. Consequently, this collaborative approach nurtures green creativity and ultimately contributes to the realization of firm PGI. Previous scholars have examined the influence of social capital theory on the relationship between SSC and a firm's green performance, leading to PGI, without considering DC theory. Given that SSC serves as a crucial starting point in social capital and DC theories [28], this study pioneers an exploration of the theoretical connections among SSC, CSC, RSC, DC, and PGI.

Secondly, while existing research by Ding [20], Jiang [131] and Annamalah, Paraman [60] has explored the direct impact of social capital on the environment, this study addresses the relationship between social capital and PGI, an aspect that has been underexamined in earlier research. By doing so, this study contributes to the literature by providing a comprehensive review of the significance of proactive green managerial practices within the manufacturing industry, highlighting their role in environmental protection and the maintenance of long-term market share. By investigating the interconnectedness between environmentally friendly management practices and green innovation for sustainability within manufacturing enterprises, we augment existing green manufacturing knowledge by introducing a fresh perspective on the theoretical understanding of sustainable innovation, extending beyond the scope of socially sustainable consumption (SSC). Moreover, the empirical evidence substantiates these theoretical perspectives, opening avenues for further exploration of sustainable innovation's implications for enterprises. This is particularly noteworthy as we have identified a non-significant relationship between SSC and PGI in the manufacturing context.

Thirdly, the examination of the individual mediating roles of CSC, RSC, and DC has often been fragmented and inconclusive. This study sought to transcend these fragmented theoretical connections between different factors and their relationships with SSC and PGI. By doing

so, we lay the groundwork for numerous future studies in the domains of green manufacturing, enterprise green innovation, and environmentally conscious competition, all geared toward environmental preservation and the attraction of eco-conscious consumers.

Fourthly, this study contributes to a comprehensive understanding of the relationship between enterprise green innovation and socially sustainable consumption (SSC) by investigating the sequential mediating influences of CSC, RSC, and DC within a single framework. Previous research has explored the influence of individual variables, such as DC, but such perspectives have been limited in scope [30,132]. No prior research has identified the effects of CSC, RSC, and DC as sequential mediators. Hence, this study bridges the gap at both the theoretical and methodological levels by employing a serial mediation research model that examines these variables' sequential impacts on the relationship between SSC and PGI. This endeavor makes a significant contribution, as it enriches our conceptual understanding of how enterprise green innovation plays a pivotal role in promoting sustainable practices and achieving environmentally friendly outcomes.

Ultimately, this study addresses existing research gaps in the field of green manufacturing, presenting a fresh theoretical perspective that integrates and synthesizes current theoretical underpinnings while providing valuable empirical evidence. Consequently, it contributes to a more holistic understanding of green manufacturing and fills the knowledge voids present in the literature.

## 5.2 Practical implications

The findings of this study have practical implications for the management of manufacturing firms. Initially, manufacturing companies adopt SSC practices to cultivate valuable social resources. However, it is important to note that SSC practices alone do not guarantee PGI. As a result, businesses are encouraged to incorporate green innovation within their SSC strategies. The SSC practices within the manufacturing industry can inspire enterprises to embrace environmentally friendly approaches to manufacturing, ultimately leading to PGI.

Manufacturing managers can play a pivotal role in shaping the external environment of their enterprises. They can enhance the frequency, quantity, and quality of connections with other enterprises through SSC, thereby fostering CSC. This, in turn, results in the development of a shared vision, culture, topics, and values among connected enterprises, forming RSC. Inter-organizational trust is cultivated through these relationships, promoting cooperation, reciprocity, and commitments between organizations. Consequently, organizations can share social capital, exchange green information and resources, support PGI, and facilitate environmental sustainability within the manufacturing industry.

Additionally, based on the outcomes of this study, it is evident that the dynamic capabilities of manufacturing enterprises are a result of comprehensive SSC practices. These practices trigger a sequence of CSC, RSC, and DC formation. Manufacturing enterprises that prioritize sustainable development reflect a commitment to environmentally friendly innovation practices. For example, these enterprises can strengthen the frequency, quality, and quantity of social interactions with other enterprises through SSC, fostering a common green culture and environmental awareness (CSC). They can also build trust with other enterprises to collaboratively address climate change (RSC). This enables them to adapt to changing business environments, tackle environmental challenges, acquire green information, knowledge, and technology, and transform externally sourced resources into their own innovative capabilities (DC). This, in turn, can lead to reduced carbon emissions and promote PGI within the manufacturing industry [22].

Furthermore, it is essential for enterprises to leverage their sensing and seizing capabilities to quickly access information related to green innovation policies, cutting-edge industry

technologies, user needs, and potential economic, social, and environmental benefits. Such access to information can stimulate and enhance enterprises' willingness to engage in PGI. Subsequently, with insights gained through sensing and seizing capabilities, enterprises can reconfigure their existing resources and capabilities into new ones suitable for PGI, thereby improving their overall innovation capacity.

CSC and RSC play important roles in shaping DC. To positively influence these factors within enterprises, manufacturing companies may consider recruiting talent or participating in conferences and social organizations related to green innovation to increase their interaction with society and the industry. SSC, as a dimension of social capital emphasizing the quantity, quality, and frequency of interactions between organizations, facilitates resource acquisition and establishes a platform for manufacturing enterprises to access resources. Consequently, active green innovation by manufacturing enterprises helps them maximize market share and gain a competitive advantage, attracting environmentally conscious customers and leading the market.

Next, some manufacturing enterprises engage in deceptive practices by falsely claiming to adopt SSC practices, which is commonly referred to as "greenwashing" [30]. Deceptive SSC practices can mislead others and create misconceptions, leading to a reluctance among enterprises to engage in CSC activities [133]. This study underscores the critical role of RSC and DC in driving enterprise green innovation and ultimately achieving PGI. It highlights that merely adopting socially sustainable consumption practices is insufficient without the presence of dynamic capabilities within the organization. Manufacturing enterprises lacking the ability to drive green innovation may encounter challenges in realizing PGI.

Therefore, it is imperative for managers to acknowledge the significance of nurturing RSC and cultivating dynamic capabilities to effectively implement sustainable practices and foster innovation in pursuit of PGI. This study serves as a reference for manufacturing sector practitioners to gain a deeper understanding of the sequential roles played by enterprise CSC, RSC, and DC in relation to SSC for achieving PGI.

## 5.3 Limitations and future research

This study is subject to several limitations that pave the way for future research. Firstly, although the results contribute significantly to the understanding of the relationship between SSC and PGI, future studies are encouraged to explore the role of DC and its sub-dimensions (i.e., sensing, seizing, and reconfiguring) [69] in conjunction with social capital, irrespective of its dimension, in the context of PGI.

Secondly, to broaden the scope of our proposed conceptual framework, future research endeavors might consider investigating absorptive capacity and knowledge creation as potential moderators that may vary firms' cognitive responses to environmental stimuli [2]. It would also be intriguing to probe further into the moderating influence of organizational DC on the relationships suggested in this study, as prior research [60] has demonstrated its impact on the interplay among SSC, RSC, and CSC.

Thirdly, while our research model provides valuable insights into PGI, social capital, and DC, we recommend that future scholars expand their research by incorporating various measurements for PGI, given the absence of a consensus on its constituent components [20,131]. In conclusion, the manufacturing industry requires a more explicit and conclusive understanding of SSC and PGI, necessitating further research on this topic in the future.

Fourthly, Given that we employed a cross-sectional design, causal relationships cannot be established. Therefore, future research could utilize a longitudinal study design to validate the serial mediation model we identified and further investigate the evolution of these

relationships over time. This would allow for a more comprehensive understanding of how social capital, dynamic capabilities, and PGI interplay and evolve within organizations across different time points. Additionally, longitudinal studies would provide valuable insights into the temporal dynamics and causality of these relationships, contributing to a deeper understanding of the mechanisms driving PGI in organizational settings.

## Supporting information

**S1 Table. Questionnaire items.**
(DOCX)

**S1 Data.**
(XLSX)

## Author Contributions

**Conceptualization:** Xinxiang Gao.

**Data curation:** Xinxiang Gao.

**Formal analysis:** Xinxiang Gao.

**Funding acquisition:** Xinxiang Gao.

**Investigation:** Xinxiang Gao.

**Methodology:** Xinxiang Gao.

**Resources:** Xinxiang Gao.

**Software:** Xinxiang Gao.

**Visualization:** Xinxiang Gao.

**Writing – original draft:** Xinxiang Gao.

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
