## [Decision Letter · Decision Letter 0]

9 Feb 2024

PONE-D-23-43928Does Structure Social Capital Lead to a Proactive Green Innovation? A Serial Mediation Model with Three MediatorsPLOS ONE

Dear Dr. Gao,

Thank you for submitting your manuscript to PLOS ONE. After careful consideration, we feel that it has merit but does not fully meet PLOS ONE’s publication criteria as it currently stands. Therefore, we invite you to submit a revised version of the manuscript that addresses the points raised during the review process.

We look forward to receiving your revised manuscript.

Kind regards,

Bablu Kumar Dhar, PhD, Post Doc

Academic Editor

PLOS ONE

Journal Requirements:

2. You indicated that ethical approval was not necessary for your study. We understand that the framework for ethical oversight requirements for studies of this type may differ depending on the setting and we would appreciate some further clarification regarding your research. 

Could you please provide further details on why your study is exempt from the need for approval and confirmation from your institutional review board or research ethics committee (e.g., in the form of a letter or email correspondence) that ethics review was not necessary for this study? 

Please include a copy of the correspondence as an ""Other"" file.

a) The name of the colleague or the details of the professional service that edited your manuscript.

b) A copy of your manuscript showing your changes by either highlighting them or using track changes (uploaded as a *supporting information* file).

c) A clean copy of the edited manuscript (uploaded as the new *manuscript* file).

b) If there are no restrictions, please upload the minimal anonymized data set necessary to replicate your study findings to a stable, public repository and provide us with the relevant URLs, DOIs, or accession numbers. For a list of recommended repositories, please see https://journals.plos.org/plosone/s/recommended-repositories. You also have the option of uploading the data as Supporting Information files, but we would recommend depositing data directly to a data repository if possible.

Reviewers' comments:

Reviewer's Responses to Questions

**Comments to the Author**

1. Is the manuscript technically sound, and do the data support the conclusions?

Reviewer #1: Yes

Reviewer #2: Yes

2. Has the statistical analysis been performed appropriately and rigorously? 

Reviewer #1: Yes

Reviewer #2: Yes

3. Have the authors made all data underlying the findings in their manuscript fully available?

Reviewer #1: Yes

Reviewer #2: Yes

4. Is the manuscript presented in an intelligible fashion and written in standard English?

Reviewer #1: Yes

Reviewer #2: Yes

5. Review Comments to the Author

Reviewer #1: After reviewing the manuscript "Does Structure Social Capital Lead to a Proactive Green Innovation? A Serial Mediation Model with Three Mediators," here's a review report with critical research questions and suggestions for additional references:

Critical Research Questions:

Theoretical Framework: How well does the manuscript integrate and build upon existing theories of social capital and dynamic capabilities to explain proactive green innovation? Is the theoretical linkage between structural, cognitive, and relational social capital clearly articulated in relation to proactive green innovation?

Methodological Rigor: Are the methods used for data collection and analysis sufficiently robust and appropriate for testing the proposed serial mediation model? How effectively does the manuscript address potential limitations associated with the cross-sectional research design and the use of structural equation modeling?

Empirical Evidence: How compelling and reliable are the empirical findings in supporting the proposed serial mediation effects among structural social capital, cognitive social capital, relational social capital, dynamic capabilities, and proactive green innovation?

Practical Implications: Does the manuscript clearly outline the practical implications of its findings for managers and practitioners in the manufacturing industry seeking to leverage social capital for green innovation?

Future Research Directions: How well does the manuscript identify and articulate avenues for future research, especially concerning the limitations of the current study and the potential for longitudinal studies to validate the findings?

Suggested References for Inclusion:

Dhar, B. K., Sarkar, S. M., & Ayittey, F. K. (2022). Impact of social responsibility disclosure between implementation of green accounting and sustainable development: A study on heavily polluting companies in Bangladesh. Corporate Social Responsibility and Environmental Management, 29(1), 71-78.

Ali, M. K., Zahoor, M. K., Saeed, A., Nosheen, S., & Thanakijsombat, T. (2023). Impact of Vertical Integration Strategies on Environmental, Social, and Governance Sustainability: Policy Implication for Oil and Gas Energy Sector. Process Integration and Optimization for Sustainability, 1-15.

Ahmed, S., Ashrafi, D. M., Paraman, P., Dhar, B. K., & Annamalah, S. (2023). Behavioural intention of consumers to use app-based shopping on green tech products in an emerging economy. International Journal of Quality & Reliability Management, (ahead-of-print).

Sundararajan, N., Habeebsheriff, H. S., Dhanabalan, K., Cong, V. H., Wong, L. S., Rajamani, R., & Dhar, B. K. (2023). Mitigating Global Challenges: Harnessing Green Synthesized Nanomaterials for Sustainable Crop Production Systems. Global Challenges, 2300187.

Ali, M. K., Zahoor, M. K., Saeed, A., Nosheen, S., & Thanakijsombat, T. (2023). Institutional and country level determinants of vertical integration: New evidence from the oil and gas industry. Resources Policy, 84, 103777.

Absar, M. M. N., Dhar, B. K., Mahmood, M., & Emran, M. (2021). Sustainability disclosures in emerging economies: Evidence from human capital disclosures on listed banks' websites in Bangladesh. Business and Society Review, 126(3), 363-378.

These references provide a broader context on green innovation, social capital, and sustainable practices in emerging economies, which could enrich the manuscript's discussion and theoretical grounding.

Reviewer #2: These questions are designed to probe deeper into the nuances of the research findings and to suggest areas for further investigation that could enrich the understanding of the dynamics between structural social capital and proactive green innovation. Please address these during your revision:

How does the cultural and institutional context of different countries or regions influence the relationship between structural social capital and proactive green innovation? This question seeks to understand if the model's applicability varies across different socio-economic environments.

In what ways do recent technological advancements and digital transformation influence the mediating roles of cognitive and relational social capital, as well as dynamic capabilities, in fostering proactive green innovation?

What are the long-term impacts of structural social capital on proactive green innovation beyond the immediate effects captured in the study?

6. PLOS authors have the option to publish the peer review history of their article (what does this mean?). If published, this will include your full peer review and any attached files.

Reviewer #1: **Yes: **Sabrina Maria Sarkar, Department of Public Policy, CIES, ISCTE-IUL, Lisbon, Portugal

Reviewer #2: No

---

## [Author Response · Author response to Decision Letter 0]

8 Mar 2024

Dear Editor

I want to express my sincere gratitude for the opportunity to submit my paper to PLOS ONE. Your guidance and support throughout the review process have been invaluable. Your expertise and dedication to fostering quality research in our field are truly commendable. I appreciate the time and effort you and your team have invested in evaluating my work and providing constructive feedback.

Thank you for your feedback and guidance. I have made the necessary adjustments to ensure that the manuscript meets PLOS ONE’s style requirements, including file naming conventions. Please find the revised version attached. If there are any further changes or adjustments needed, please do not hesitate to let me know.

2. You indicated that ethical approval was not necessary for your study. We understand that the framework for ethical oversight requirements for studies of this type may differ depending on the setting and we would appreciate some further clarification regarding your research. 

Could you please provide further details on why your study is exempt from the need for approval and confirmation from your institutional review board or research ethics committee (e.g., in the form of a letter or email correspondence) that ethics review was not necessary for this study? 

Please include a copy of the correspondence as an ""Other"" file.

Thank you for your feedback regarding the need for ethical approval for our study.

We understand the importance of ensuring compliance with ethical standards in research, and we appreciate the opportunity to provide further clarification.

Attached, please find the approval form our institutional SEGi Research Ethics Committee (REC) confirming (Ethics Approval Number: SEGiEC/SR/GSB/161/2023-2024). This approval was obtained after careful consideration of the study design, methodology, and potential ethical implications.

We trust that this documentation addresses your concerns regarding the ethical oversight of our research. If you require any additional information or clarification, please do not hesitate to let us know.

Thank you for your advice. I have already enlisted the help of professional experts to proofread and edit the manuscript for language usage, spelling, and grammar. I believe this will greatly enhance the quality of the manuscript.

b) If there are no restrictions, please upload the minimal anonymized data set necessary to replicate your study findings to a stable, public repository and provide us with the relevant URLs, DOIs, or accession numbers. For a list of recommended repositories, please see https://journals.plos.org/plosone/s/recommended-repositories. You also have the option of uploading the data as Supporting Information files, but we would recommend depositing data directly to a data repository if possible.

Thank you. I acknowledge the importance of data transparency and am committed to making the data set used in this study publicly available.

I have prepared the minimal anonymized data set necessary to replicate the study findings and I chose to submit the data as supporting files because I'm not familiar with using data repositories. I've attempted to use them several times, but unfortunately, without success. I hope you understand. If my choice has caused any inconvenience in your work, I apologize. Thank you for your understanding.

Please update the Data Availability statement accordingly, and let me know if there are any additional steps required from my end.

Thank you once again for your commitment to advancing scholarly knowledge and for considering my manuscript for publication in PLOS ONE. We look forward to the next steps in the review process.

Warm regards,

Xinxiang Gao

Comments to the Author

5. Review Comments to the Author

Reviewer #1: After reviewing the manuscript "Does Structure Social Capital Lead to a Proactive Green Innovation? A Serial Mediation Model with Three Mediators," here's a review report with critical research questions and suggestions for additional references:

Dear Reviewer,

I would like to extend my sincere gratitude for your valuable questions and suggestions on my paper. Your review not only provides crucial feedback but also guides me in refining my work. Your expertise is greatly appreciated, and I will carefully consider your insights as I continue to develop my research.

Q1: Theoretical Framework: How well does the manuscript integrate and build upon existing theories of social capital and dynamic capabilities to explain proactive green innovation? Is the theoretical linkage between structural, cognitive, and relational social capital clearly articulated in relation to proactive green innovation?

Our study indeed aims to integrate theories of social capital and dynamic capabilities to explain proactive green innovation. We have extensively discussed these theories in the literature review and proposed a framework aimed at elucidating their roles in the process of green innovation.The manuscript integrates and builds upon existing theories of social capital and dynamic capabilities to explain proactive green innovation. And the theoretical linkage between structural, cognitive, and relational social capital clearly articulated in relation to proactive green innovation. 

Firstly, Bourdieu and Richardson (1986) formally introduced the concept of social capital, which encompasses all resources embedded within a social network characterized by behavioral norms and close interpersonal connections. Social capital is instrumental in assisting individuals and organizations in achieving specific predetermined objectives (Inkpen & Tsang, 2005). In essence, social capital theory views social networks as vital conduits for individuals and entities to access information and resources (Appiah & Obey, 2023; Carey et al., 2011; Putnam, 2015). It posits that the collective resources offered by network relationships foster mutual trust among network members across various domains, making it a valuable asset for individuals to leverage (Lin, 2017). Therefore, this study draws upon social capital theory as one of its foundational frameworks to elucidate the impact of external social capital on the realization of proactive green innovation goals within enterprises. 

Secondly, Previous research has already explored the influence of external social capital on various fronts. In the domain of social responsibility, Gao et al. (2021) contend that robust social norms and dense social networks, cultivated by strong external social capital, serve to curtail unethical corporate behavior and enhance resource utilization by enterprises. In the field of strategic management, external social capital aids enterprises in acquiring unique abilities and resources that are challenging for competitors to replicate, thereby strengthening overall competitiveness (Annamalah et al., 2023). Lyu et al. (2022) posit that external social capital has an indirect effect on enterprise innovation by shaping knowledge acquisition. From an innovator's perspective, Zhang et al. (2022) suggest that green innovation within enterprises can be influenced by interactions with a wide array of primary and secondary stakeholders. The engagement between enterprises and external network participants fosters an environment conducive to green innovation (Ullah et al., 2022), thereby enhancing overall green innovation efforts within the enterprise (Ding, 2022; Dong et al., 2022). For the purpose of this study, we adopted the framework proposed by Nahapiet and Ghoshal (1998), which categorizes external social capital into three components: structural social capital, cognitive social capital, and relational social capital (Al-Omoush et al., 2022). 

Thirdly, According to Teece et al. (1997), organizations must possess the capability to identify opportunities and threats, acquire valuable resources, and adapt both external and internal resources to navigate the complex and ever-changing business environment (Ambrosini & Bowman, 2009; Feng et al., 2023; Teece, 2007; Zahra & George, 2002). This concept is referred to as dynamic capability, which can be further subdivided into sensing capability, seizing capability, and reconfiguring capability (Ghosh et al., 2022). Dynamic capability theory has found application in various management research fields (Feng et al., 2023; Fredrich et al., 2022; C. Wang et al., 2023). Liao et al. (2009) and Su et al. (2022) suggest that enterprise innovation is realized through the effective utilization of resources, facilitated by dynamic capabilities, while Moroni et al. (2022) argue that dynamic capability positively influences enterprise innovation. Feng et al. (2023) proposed that innovation is positively affected by dynamic capability, although this relationship is often subject to negative regulation by environmental uncertainty (Barreto, 2010). 

Fourthly, Within the literature exploring the relationship between dynamic capability and structural social capital, dynamic capability is frequently seen as a facilitator for integrating external knowledge and resources to help enterprises attain their predefined objectives (Huang et al., 2023; Jiang et al., 2020; Von Briel et al., 2019). Qiu et al. (2020) assert that dynamic capability plays an intermediary role in the connection between green innovation and competitive advantage. In general, dynamic capability is regarded as the primary source of an enterprise's competitive advantage. Enterprises that fail to adapt to their environment often lose a significant portion of their competitive edge due to a lack of dynamic capability (Akpan et al., 2021; Bornay-Barrachina et al., 2023). Grounded in the theoretical perspective of dynamic capability, the unique ability of organizations to perceive and effectively integrate information and resources from both internal and external sources becomes the cornerstone for achieving organizational innovation objectives and sustainable competitive advantages (Bernal-Torres et al., 2023). Therefore, this study incorporated dynamic capability theory as one of its foundational frameworks to elucidate how firm dynamic capabilities drive the realization of proactive green innovation within an enterprise. 

Fifthly, Lyu (2023) pointed out that social capital can influence enterprise innovation through dynamic capabilities. Additionally, in their study, Huang and Li (2017) contended that dynamic capability plays a vital role in driving green innovation, as it helps enterprises navigate environmental changes, allowing them to identify opportunities for green innovation in the market (Bernal-Torres et al., 2023). Notably, the mechanism by which social capital affects enterprise proactive green innovation is indirect, relying on the cascading effects of different types of social capital. Structural social capital, rooted in inter-organizational interactions, directly impacts cognitive social capital by stimulating the activities of various actors (Ding, 2022). It serves as a prerequisite for cognitive social capital, which encompasses shared goals, visions, values, culture, and similar factors that facilitate communication and mutual encouragement among various actors, thus establishing the cognitive conditions for the development of inter-organizational trust, cooperation, and reciprocity relationships. Moreover, structural social capital enhances the exchange of resources and the development of sensing, acquiring, and transforming capabilities between green business partners, thereby improving the effectiveness of dynamic capability (Akpan et al., 2021; Bornay-Barrachina et al., 2023). In addition, relational social capital influences the link between cognitive social capital and dynamic capability (Bernal-Torres et al., 2023; Monteiro et al., 2017), and may serve as a sequential mediator in the effectiveness of dynamic capability. Through the reciprocal influence among different dimensions of social capital, companies can cultivate genuine dynamic capabilities, which provide the resource foundation for the development and enhancement of enterprise proactive green innovation. Therefore, the cumulative effects of structural social capital, cognitive social capital, and relational social capital on dynamic capabilities ultimately impact enterprise proactive green innovation (Akpan et al., 2021; Singh et al., 2022).

Q2: Methodological Rigor: Are the methods used for data collection and analysis sufficiently robust and appropriate for testing the proposed serial mediation model? How effectively does the manuscript address potential limitations associated with the cross-sectional research design and the use of structural equation modelling?

Thank you for your inquiry regarding the adequacy of our data collection and analysis methods, as well as the manuscript's handling of potential limitations associated with the cross-sectional research design and the use of structural equation modeling (SEM).

Data Collection and Analysis Methods: The data collection method employed in our study involved purposive sampling and offline surveys, which allowed us to gather comprehensive and relevant data for testing the proposed serial mediation model. We believe this method is robust and appropriate as it enabled us to capture the necessary variables and relationships pertinent to our research objectives. Regarding data analysis, we utilized structural equation modeling (SEM) to test the hypothesized relationships among the variables. SEM offers several advantages, including its ability to analyze complex models and account for measurement error. We employed state-of-the-art statistical techniques and software to ensure the accuracy and reliability of our results.

Addressing Potential Limitations: We acknowledge that the cross-sectional research design inherently limits our ability to establish causality. However, we addressed this limitation by incorporating theoretical rationale and prior empirical evidence to support the proposed serial mediation model. Additionally, we discussed the implications of our findings in light of the cross-sectional nature of the data. Concerning the use of structural equation modeling, we took several steps to mitigate potential limitations, such as model misspecification and common method bias. For example, we conducted sensitivity analyses, including alternative model specifications, to ensure the robustness of our results. Moreover, we employed techniques such as marker variable analysis to assess and control for common method bias.

In summary, we believe that the methods used for data collection and analysis are sufficiently robust and appropriate for testing the proposed serial mediation model. Furthermore, we have taken proactive measures to address potential limitations associated with the cross-sectional research design and the use of structural equation modeling in our manuscript.

Q3: Empirical Evidence: How compelling and reliable are the empirical findings in supporting the proposed serial mediation effects among structural social capital, cognitive social capital, relational social capital, dynamic capabilities, and proactive green innovation?

Thank you for your insightful questions regarding the empirical findings of our study. We appreciate the opportunity to delve deeper into the robustness and implications of our results.

In evaluating the empirical findings, we find them to be compelling and reliable in supporting the proposed serial mediation effects among structural, cognitive, and relational social capital, dynamic capabilities, and proactive green innovation. The statistical analysis, including structural equation modeling, yielded significant relationships between the variables, providing strong support for our theoretical framework.

While the findings offer compelling evidence for the hypothesized relationships, we acknowledge several limitations associated with the cross-sectional research design and the use of structural equation modeling. Despite these limitations, we have taken measures to address potential concerns by employing rigorous data collection methods and sensitivity analyses to ensure the robustness of our results. Additionally, we have provided thorough discussions on the limitations and implications of our findings in the manuscript.

Moving forward, future research could benefit from longitudinal studies or experimental designs to establish causality and further validate the serial mediation effects proposed in our model. Additionally, exploring alternative methodologies or incorporating additional control variables could enhance the reliability and generalizability of our findings.

Q4: Practical Implications: Does the manuscript clearly outline the practical implications of its findings for managers and practitioners in the manufacturing industry seeking to leverage social capital for green innovation?

The manuscript clearly outlines the practical implications of its findings for managers and practitioners in the manufacturing industry seeking to leverage social capital for green innovation.

The findings of this study have practical implications for the management of manufacturing firms. Initially, manufacturing companies adopt structural social capital practices to cultivate valuable social resources. However, it is important to note that structural social capital practices alone do not guarantee proactive green innovation. As a result, businesses are encouraged to incorporate green innovation within their structural social capital strategies. The structural social capital practices within the manufacturing industry can inspire enterprises to embrace environmentally friendly approaches to manufacturing, ultimately leading to proactive green innovation.

Manufacturing managers can play a pivotal role in shaping the external environment of their enterprises. They can enhance the frequency, quantity, and quality of connections with other enterprises through structural social capital, thereby fostering cognitive social capital. This, in turn, results in the development of a shared vision, culture, topics, and values among connected enterprises, forming relational social capital. Inter-organizational trust is cultivated through these relationships, promoting cooperation, reciprocity, and commitments between organizations. Consequently, organizations can share social capital, exchange green information and resources, support proactive green innovation, and facilitate environmental sustainability within the manufacturing industry.

Additionally, based on the outcomes of this study, it is evident that the dynamic capabilities of manufacturing enterprises are a result of comprehensive structural social capital practices. These practices trigger a sequence of cognitive social capital, relational social capital, and dynamic capability formation. Manufacturing enterprises that prioritize sustainable development reflect a commitment to environmentally friendly innovation practices. For example, these enterprises can strengthen the frequency, quality, and quantity of social interactions with other enterprises through structural social capital, fostering a common green culture and environmental awareness (cognitive social capital). They can also build trust with other enterprises to collaboratively address climate change (relational social capital). This enables them to adapt to changing business environments, tackle environmental challenges, acquire green information, knowledge, and technology, and transform externally sourced resources into their own innovative capabilities (dynamic capability). This, in turn, can lead to reduced carbon emissions and promote proactive green innovation within the manufacturing industry (Bataineh et al., 2023).

Furthermore, it is essential for enterprises to leverage their sensing and seizing capabilities to quickly access information related to green innovation policies, cutting-edge industry technologies, user needs, and potential economic, social, and environmental benefits. Such access to information can stimulate and enhance enterprises' willingness to engage in proactive green innovation. Subsequently, with insights gained through sensing and seizing capabilities, enterprises can reconfigure their existing resources and capabilities into new ones suitable for proactive green innovation, thereby improving their overall innovation capacity.

Cognitive social capital and relational social capital play important roles in shaping dynamic capability. To positively influence these factors within enterprises, manufacturing companies may consider recruiting talent or participating in conferences and social organizations related to green innovation to increase their interaction with society and the industry. Structural social capital, as a dimension of social capital emphasizing the quantity, quality, and frequency of interactions between organizations, facilitates resource acquisition and establishes a platform for manufacturing enterprises to access resources. Consequently, active green innovation by manufacturing enterprises helps them maximize market share and gain a competitive advantage, attracting environmentally conscious customers and leading the market.

Next, some manufacturing enterprises engage in deceptive practices by falsely claiming to adopt structural social capital practices, which is commonly referred to as "greenwashing" (Appiah & Obey, 2023). Deceptive structural social capital practices can mislead others and create misconceptions, leading to a reluctance among enterprises to engage in cognitive social capital activities (Al-Omoush et al., 2020). This study underscores the critical role of relational social capital and dynamic capability in driving enterprise green innovation and ultimately achieving proactive green innovation. It highlights that merely adopting socially sustainable consumption practices is insufficient without the presence of dynamic capabilities within the organization. Manufacturing enterprises lacking the ability to drive green innovation may encounter challenges in realizing proactive green innovation.

Therefore, it is imperative for managers to acknowledge the significance of nurturing relational social capital and cultivating dynamic capabilities to effectively implement sustainable practices and foster innovation in pursuit of proactive green innovation. This study serves as a reference for manufacturing sector practitioners to gain a deeper understanding of the sequential roles played by enterprise cognitive social capital, relational social capital, and dynamic capability in relation to structural social capital for achieving proactive green innovation.

Q5：Future Research Directions: How well does the manuscript identify and articulate avenues for future research, especially concerning the limitations of the current study and the potential for longitudinal studies to validate the findings?

The manuscript adeptly identifies and articulates several avenues for future research, particularly considering the limitations inherent in the current study design. It acknowledges that while the research contributes valuable insights into the relationship between social capital, dynamic capabilities, and proactive green innovation, certain constraints necessitate further exploration.

Firstly, although the results contribute significantly to the understanding of the relationship between structural social capital and proactive green innovation, future studies are encouraged to explore the role of dynamic capability and its sub-dimensions (i.e., sensing, seizing, and reconfiguring) (Fredrich et al., 2022) in conjunction with social capital, irrespective of its dimension, in the context of proactive green innovation.

Secondly, to broaden the scope of our proposed conceptual framework, future research endeavors might consider investigating absorptive capacity and knowledge creation as potential moderators that may vary firms’ cognitive responses to environmental stimuli (Baste & Watson, 2022). It would also be intriguing to probe further into the moderating influence of organizational dynamic capability on the relationships suggested in this study, as prior research (Annamalah et al., 2023) has demonstrated its impact on the interplay among structural social capital, relational social capital, and cognitive social capital.

Thirdly, while our research model provides valuable insights into proactive green innovation, social capital, and dynamic capability, we recommend that future scholars expand their research by incorporating various measurements for proactive green innovation, given the absence of a consensus on its constituent components (Ding, 2022; Jiang, 2022). In conclusion, the manufacturing industry requires a more explicit and conclusive understanding of structural social capital and proactive green innovation, necessitating further research on this topic in the future.

Fourthly, Given that we employed a cross-sectional design, causal relationships cannot be established. Therefore, future research could utilize a longitudinal study design to validate the serial mediation model we identified and further investigate the evolution of these relationships over time. This would allow for a more comprehensive understanding of how social capital, dynamic capabilities, and proactive green innovation interplay and evolve within organizations across different time points. Additionally, longitudinal studies would provide valuable insights into the temporal dynamics and causality of these relationships, contributing to a deeper understanding of the mechanisms driving proactive green innovation in organizational settings.

Suggested References for Inclusion:

1. Dhar, B. K., Sarkar, S. M., & Ayittey, F. K. (2022). Impact of social responsibility disclosure between implementation of green accounting and sustainable development: A study on heavily polluting companies in Bangladesh. Corporate Social Responsibility and Environmental Management, 29(1), 71-78. 

2. Ali, M. K., Zahoor, M. K., Saeed, A., Nosheen, S., & Thanakijsombat, T. (2023). Impact of Vertical Integration Strategies on Environmental, Social, and Governance Sustainability: Policy Implication for Oil and Gas Energy Sector. Process Integration and Optimization for Sustainability, 1-15.

3. Ahmed, S., Ashrafi, D. M., Paraman, P., Dhar, B. K., & Annamalah, S. (2023). Behavioural intention of consumers to use app-based shopping on green tech products in an emerging economy. International Journal of Quality & Reliability Management, (ahead-of-print).

4. Sundararajan, N., Habeebsheriff, H. S., Dhanabalan, K., Cong, V. H., Wong, L. S., Rajamani, R., & Dhar, B. K. (2023). Mitigating Global Challenges: Harnessing Green Synthesized Nanomaterials for Sustainable Crop Production Systems. Global Challenges, 2300187.

5. Ali, M. K., Zahoor, M. K., Saeed, A., Nosheen, S., & Thanakijsombat, T. (2023). Institutional and country level determinants of vertical integration: New evidence from the oil and gas industry. Resources Policy, 84, 103777.

6. Absar, M. M. N., Dhar, B. K., Mahmood, M., & Emran, M. (2021). Sustainability disclosures in emerging economies: Evidence from human capital disclosures on listed banks' websites in Bangladesh. Business and Society Review, 126(3), 363-378.

These references provide a broader context on green innovation, social capital, and sustainable practices in emerging economies, which could enrich the manuscript's discussion and theoretical grounding.

We sincerely appreciate the valuable comments. We have checked the literature carefully and added references on 1&2&3 into the Theoretical focus part in the revised manuscript， and added references on 4&5&6 into the Introduction part in the revised manuscript.

1 Dhar et al. (2022) discovered that the quality of social responsibility information disclosure can be positively adjusted to the relationship between the implementation of green accounting and the sustainable development capabilities of heavily polluting companies.

2&3 Ali et al. (2023a) and Ali et al. (2023b) that the petroleum industry continuously adapts to future scenarios by adopting innovative technologies while seeking to mitigate adverse impacts on society and addressing risks associated with climate change to promote sustainable development.

4 Ahmed et al. (2023) that the innovative behavior of consumers in emerging economies using applications to purchase green technology products can promote environmental conservation.

5 Sundararajan et al. (2024) Innovative research and development of green synthetic nanomaterials can promote sustainable crop production systems, paving the way for future sustainable crop production systems.

6 Absar et al. (2021) Sustainability disclosure in emerging markets benefits the development of the green manufacturing industry.

Once again, we appreciate your thoughtful questions and feedback, which have contributed to the refinement of our study's empirical foundation.

Thank you once again for your thoughtful review and contribution.

Best regards

Reviewer #2: These questions are designed to probe deeper into the nuances of the research findings and to suggest areas for further investigation that could enrich the understanding of the dynamics between structural social capital and proactive green innovation. Please address these during your revision:

Dear Reviewer,

I would like to extend my sincere gratitude for your valuable questions and suggestions on my paper. Your review not only provides crucial feedback but also guides me in refining my work. Your expertise is greatly appreciated, and I will carefully consider your insights as I continue to develop my research.

How does the cultural and institutional context of different countries or regions influence the relationship between structural social capital and proactive green innovation? This question seeks to understand if the model's applicability varies across different socio-economic environments.

The influence of cultural and institutional contexts on the relationship between structural social capital and proactive green innovation is a crucial aspect of my research. China’s unique cultural values and institutional framework can significantly shape the dynamics of social capital formation and its impact on environmental innovation (Lin et al., 2014).

In China, traditional cultural values emphasizing collective harmony and social cohesion often contribute to the formation of strong social networks and trust within communities(Wang & Liu, 2010). These social ties can facilitate knowledge sharing, collaboration, and resource mobilization, which are essential for fostering proactive green innovation initiatives.

Moreover, China’s institutional context, including government policies, regulatory frameworks, and market conditions, plays a pivotal role in incentivizing or constraining green innovation efforts (Wang et al., 2022). For example, the Chinese government's initiatives to promote sustainable development, such as the Green Development Strategy and the Belt and Road Initiative, can provide both financial and regulatory support for green innovation projects(Nedopil, 2022; S. Wang et al., 2023; Xu et al., 2022).

However, it's essential to recognize that China's institutional landscape is complex and dynamic, with varying degrees of centralization, bureaucratic structures, and enforcement mechanisms across different regions and sectors. These institutional factors can shape the accessibility of resources, the level of regulatory compliance, and the degree of collaboration among stakeholders, all of which can influence the effectiveness of social capital in driving green innovation.

Therefore, while the conceptual model of structural social capital and proactive green innovation may offer valuable insights, its applicability in China needs to be carefully examined within the specific cultural and institutional context of the country. By considering these contextual factors, my research aims to provide a nuanced understanding of how social capital dynamics contribute to green innovation in China and how policymakers and practitioners can leverage them to promote sustainable development.

In what ways do recent technological advancements and digital transformation influence the mediating roles of cognitive and relational social capital, as well as dynamic capabilities, in fostering proactive green innovation?

Recent technological advancements and digital transformation have profoundly reshaped the landscape within which cognitive and relational social capital, as well as dynamic capabilities, operate to foster proactive green innovation(Chen et al., 2022; Lyu et al., 2022). While not directly variables in our study, these advancements serve as contextual factors that influence the mechanisms through which social capital and dynamic capabilities contribute to green innovation.

Technological progress has altered the way individuals and organizations interact, communicate, and collaborate, thus affecting the development and utilization of social capital. Digital platforms, such as online forums and social media, facilitate the exchange of information and ideas, enhancing cognitive social capital by broadening access to knowledge and expertise related to green innovation. Likewise, digital tools enable the formation of virtual networks and communities, expanding relational social capital by connecting stakeholders across geographical boundaries and fostering collaboration on environmental initiatives(Ghosh et al., 2022; Yuan & Pan, 2023).

Furthermore, digital transformation has implications for the development of dynamic capabilities necessary for proactive green innovation. Technologies like big data analytics, IoT, and cloud computing enable firms to gather real-time environmental data, analyze trends, and develop responsive strategies. These capabilities empower organizations to adapt to changing market conditions, regulatory requirements, and stakeholder expectations, thereby facilitating the implementation of innovative green practices (Pedota, 2023).

Although not directly measured in my study, the influence of technological advancements and digital transformation on the mediating roles of cognitive and relational social capital, as well as dynamic capabilities, is critical to understanding the broader context within which green innovation occurs. By acknowledging these contextual factors, my research seeks to provide insights into how organizations leverage social capital and dynamic capabilities in response to the evolving technological landscape to drive proactive green innovation.

What are the long-term impacts of structural social capital on proactive green innovation beyond the immediate effects captured in the study?

The long-term impacts of structural social capital on proactive green innovation extend beyond the immediate effects captured in my study, influencing various aspects of organizational behavior and environmental sustainability initiatives over time.

Firstly, structural social capital fosters the development of enduring relationships and networks among stakeholders, which can serve as foundations for sustained collaboration and knowledge exchange in the pursuit of green innovation. These networks facilitate the continuous flow of information, resources, and support, enabling organizations to adapt to evolving environmental challenges and opportunities.

Moreover, structural social capital contributes to the accumulation of social norms, trust, and reciprocity within communities, which are essential for building resilient systems of environmental governance and collective action. Over the long term, these norms and values promote a culture of sustainability, encouraging individuals and organizations to prioritize green innovation and adopt environmentally responsible practices as integral components of their operations.

Furthermore, the influence of structural social capital on proactive green innovation extends beyond organizational boundaries to shape broader societal attitudes and policies towards environmental sustainability. As networks of social capital expand and strengthen, they can exert influence on decision-making processes, advocacy efforts, and policy development initiatives, leading to systemic changes in favor of green innovation and sustainable development.

However, it's essential to recognize that the long-term impacts of structural social capital on proactive green innovation may vary depending on contextual factors such as cultural norms, institutional arrangements, and market conditions. Therefore, future research should explore how these contextual factors interact with social capital dynamics to shape the trajectory of green innovation outcomes over time.

Thank you once again for your thoughtful review and contribution.

Best regards

References

Absar, M. M. N., Dhar, B. K., Mahmood, M., & Emran, M. (2021). Sustainability disclosures in emerging economies: Evidence from human capital disclosures on listed banks' websites in Bangladesh. Business and Society Review, 126(3), 363-378. https://doi.org/10.1111/basr.12242

Ahmed, S., Ashrafi, D. M., Paraman, P., Dhar, B. K., & Annamalah, S. (2023). Behavioural intention of consumers to use app-based shopping on green tech products in an emerging economy. International Journal of Quality & Reliability Management. https://doi.org/10.1108/ijqrm-05-2023-0164

Akpan, E. E., Johnny, E., & Sylva, W. (2021). Dynamic Capabilities and Organizational Resilience of Manufacturing Firms in Nigeria. Vision: The Journal of Business Perspective, 26(1), 48-64. https://doi.org/10.1177/0972262920984545

Al-Omoush, K. S., Ribeiro-Navarrete, S., Lassala, C., & Skare, M. (2022). Networking and knowledge creation: Social capital and collaborative innovation in responding to the COVID-19 crisis. Journal of Innovation & Knowledge, 7(2). https://doi.org/10.1016/j.jik.2022.100181

Al-Omoush, K. S., Simón-Moya, V., & Sendra-García, J. (2020). The impact of social capital and collaborative knowledge creation on e-business proactiveness and organizational agility in responding to the COVID-19 crisis. Journal of Innovation & Knowledge, 5(4), 279-288. https://doi.org/10.1016/j.jik.2020.10.002

Ali, M. K., Zahoor, M. K., Saeed, A., Nosheen, S., & Thanakijsombat, T. (2023a). Impact of Vertical Integration Strategies on Environmental, Social, and Governance Sustainability: Policy Implication for Oil and Gas Energy Sector. Process Integration and Optimization for Sustainability. https://doi.org/10.1007/s41660-023-00375-2

Ali, M. K., Zahoor, M. K., Saeed, A., Nosheen, S., & Thanakijsombat, T. (2023b). Institutional and country level determinants of vertical integration: New evidence from the oil and gas industry. Resources Policy, 84. https://doi.org/10.1016/j.resourpol.2023.103777

Ambrosini, V., & Bowman, C. (2009). What are dynamic capabilities and are they a useful construct in strategic management? International Journal of Management Reviews, 11(1), 29-49. https://doi.org/10.1111/j.1468-2370.2008.00251.x

Annamalah, S., Paraman, P., Ahmed, S., Dass, R., Sentosa, I., Pertheban, T. R., Shamsudin, F., Kadir, B., Aravindan, K. L., Raman, M., Hoo, W. C., & Singh, P. (2023). The role of open innovation and a normalizing mechanism of social capital in the tourism industry. Journal of Open Innovation: Technology, Market, and Complexity, 9(2). https://doi.org/10.1016/j.joitmc.2023.100056

Appiah, L. O., & Obey, V. Q. (2023). Social capital, joint knowledge creation and relationship performance in buyer-supplier relationships. Supply Chain Forum: An International Journal, 24(2), 217-232. https://doi.org/10.1080/16258312.2023.2183709

Barreto, I. (2010). Dynamic capabilities: A review of past research and an agenda for the future. Journal of management, 36(1), 256-280. https://doi.org/10.1177/0149206309350776

Baste, I. A., & Watson, R. T. (2022). Tackling the climate, biodiversity and pollution emergencies by making peace with nature 50 years after the Stockholm Conference. Global Environmental Change, 73, 102466. https://doi.org/10.1016/j.gloenvcha.2022.102466

Bataineh, M. J., Sánchez‐Sellero, P., & Ayad, F. (2023). Green is the new black: How research and development and green innovation provide businesses a competitive edge. Business Strategy and the Environment. https://doi.org/10.1002/bse.3533

Bernal-Torres, C. A., Amaya, N., Gómez-Santos, L., Mojica-Macias, J. P., & Sierra-Parra, D. (2023). Interrelation Between the Dynamic Capabilities of Knowledge Management, Learning, Adaptation, with Innovation in Medium and Large Companies in an Emerging Economy in Times of Pandemic. Global Business Review. https://doi.org/10.1177/09721509221146412

Bornay-Barrachina, M., López-Cabrales, A. l., & Salas-Vallina, A. s. (2023). Sensing, seizing, and reconfiguring dynamic capabilities in innovative firms: Why does strategic leadership make a difference? BRQ Business Research Quarterly. https://doi.org/10.1177/23409444231185790

Bourdieu, P., & Richardson, J. G. (1986). Handbook of Theory and Research for the Sociology of Education. The forms of capital, 241, 258. 

Carey, S., Lawson, B., & Krause, D. R. (2011). Social capital configuration, legal bonds and performance in buyer–supplier relationships. Journal of Operations Management, 29(4), 277-288. https://doi.org/10.1016/j.jom.2010.08.003

Chen, N., Sun, D., & Chen, J. (2022). Digital transformation, labour share, and industrial heterogeneity. Journal of Innovation & Knowledge, 7(2). https://doi.org/10.1016/j.jik.2022.100173

Dhar, B. K., Sarkar, S. M., & Ayittey, F. K. (2022). Impact of social responsibility disclosure between implementation of green accounting and sustainable development: A study on heavily polluting companies in Bangladesh. Corporate Social Responsibility and Environmental Management, 29(1), 71-78. https://doi.org/10.1002/csr.2174

Ding, Y. (2022). Correlation Analysis Model of Social Capital and Innovation Performance Based on Knowledge Mapping [Article]. Computational Intelligence and Neuroscience, 2022, Article 2138200. https://doi.org/10.1155/2022/2138200

Dong, Q., Wu, Y., Lin, H., Sun, Z., & Liang, R. (2022). Fostering green innovation for corporate competitive advantages in big data era: The role of institutional benefits. Technology Analysis & Strategic Management, 1-14. https://doi.org/10.1080/09537325.2022.2026321

Feng, X., Zibibula, M., & Wei, C. (2023). Evaluation of dynamic technological innovation capability in high-tech enterprises based on pythagorean fuzzy LBWA and MULTIMOORA. Journal of Intelligent & Fuzzy Systems(Preprint), 1-23. https://doi.org/10.3233/JIFS-222965

Fredrich, V., Gudergan, S., & Bouncken, R. B. (2022). Dynamic capabilities, internationalization and growth of small-and medium-sized enterprises: The roles of research and development intensity and collaborative intensity. Management International Review, 62(4), 611-642. 

Gao, Z., Li, L., & Lu, L. Y. (2021). Social capital and managers’ use of corporate resources. Journal of business ethics, 168, 593-613. 

Ghosh, S., Hughes, M., Hodgkinson, I., & Hughes, P. (2022). Digital transformation of industrial businesses: A dynamic capability approach. Technovation, 113. https://doi.org/10.1016/j.technovation.2021.102414

Huang, J.-W., & Li, Y.-H. (2017). Green innovation and performance: The view of organizational capability and social reciprocity. Journal of business ethics, 145, 309-324. 

Huang, J. Y. H., Jiang, R., & Chang, J. Y. T. (2023). The Effects of Transformational and Adaptive Leadership on Dynamic Capabilities: Digital Transformation Projects. Project Management Journal, 54(4), 428-446. https://doi.org/10.1177/87569728231165896

Inkpen, A. C., & Tsang, E. W. (2005). Social capital, networks, and knowledge transfer. Academy of Management Review, 30(1), 146-165. https://doi.org/10.5465/amr.2005.15281445

Jiang, L. L., Yuanjian. Zhang, Xinlei. (2022). Research on the Influence Mechanism of International Intellectual Capital on Corporate Social Performance from the Perspective of Dynamic Capability. Forum on Science and Technology in China, No.316(08), 119-127. https://doi.org/10.13580/j.cnki.fstc.2022.08.006

Jiang, W., Mavondo, F., & Zhao, W. (2020). The impact of business networks on dynamic capabilities and product innovation: The moderating role of strategic orientation. Asia Pacific Journal of Management, 37, 1239-1266. https://doi.org/10.1007/s10490-018-9628-2

Liao, J., Kickul, J. R., & Ma, H. (2009). Organizational dynamic capability and innovation: An empirical examination of internet firms. Journal of small business management, 47(3), 263-286. 

Lin, H., Zeng, S., Ma, H., Qi, G., & Tam, V. W. (2014). Can political capital drive corporate green innovation? Lessons from China. Journal of Cleaner Production, 64, 63-72. 

Lin, N. (2017). Building a network theory of social capital. Social capital, 3-28. 

Lyu, C., Peng, C., Yang, H., Li, H., & Gu, X. (2022). Social capital and innovation performance of digital firms: Serial mediation effect of cross-border knowledge search and absorptive capacity. Journal of Innovation & Knowledge, 7(2), 100187. https://doi.org/10.1016/j.jik.2022.100187

Lyu, K. (2023). Social Capital and Self-Employment Dynamics in China. The Chinese Economy, 1-27. https://doi.org/10.1080/10971475.2023.2227028

Monteiro, A. P., Soares, A. M., & Rua, O. L. (2017). Linking intangible resources and export performance. Baltic Journal of Management, 12(3), 329-347. https://doi.org/10.1108/bjm-05-2016-0097

Nahapiet, J., & Ghoshal, S. (1998). Social capital, intellectual capital, and the organizational advantage. Academy of Management Review, 23(2), 242-266. https://doi.org/10.5465/amr.1998.533225

Nedopil, C. (2022). Green finance for soft power: An analysis of China's green policy signals and investments in the Belt and Road Initiative. Environmental Policy and Governance, 32(2), 85-97. 

Pedota, M. (2023). Big data and dynamic capabilities in the digital revolution: The hidden role of source variety. Research policy, 52(7). https://doi.org/10.1016/j.respol.2023.104812

Putnam, R. D. (2015). Bowling alone: America’s declining social capital. In The city reader (pp. 188-196). Routledge. 

Qiu, L., Jie, X., Wang, Y., & Zhao, M. (2020). Green product innovation, green dynamic capability, and competitive advantage: Evidence from Chinese manufacturing enterprises. Corporate Social Responsibility and Environmental Management, 27(1), 146-165. https://doi.org/10.1002/csr.1780

Singh, S. K., Del Giudice, M., Chiappetta Jabbour, C. J., Latan, H., & Sohal, A. S. (2022). Stakeholder pressure, green innovation, and performance in small and medium‐sized enterprises: The role of green dynamic capabilities. Business Strategy and the Environment, 31(1), 500-514. https://doi.org/10.1002/bse.2906

Su, H., Qu, X., Tian, S., Ma, Q., Li, L., & Chen, Y. (2022). Artificial intelligence empowerment: The impact of research and development investment on green radical innovation in high‐tech enterprises. Systems Research and Behavioral Science, 39(3), 489-502. https://doi.org/10.1002/sres.2853

Sundararajan, N., Habeebsheriff, H. S., Dhanabalan, K., Cong, V. H., Wong, L. S., Rajamani, R., & Dhar, B. K. (2024). Mitigating Global Challenges: Harnessing Green Synthesized Nanomaterials for Sustainable Crop Production Systems. Glob Chall, 8(1), 2300187. https://doi.org/10.1002/gch2.202300187

Teece, D. J. (2007). Explicating dynamic capabilities: the nature and microfoundations of (sustainable) enterprise performance. Strategic Management Journal, 28(13), 1319-1350. https://doi.org/10.1002/smj.640

Teece, D. J., Pisano, G., & Shuen, A. (1997). Dynamic capabilities and strategic management. Strategic Management Journal, 18(7), 509-533. https://doi.org/10.1002/(SICI)1097-0266(199708)18:7<509::AID-SMJ882>3.0.CO;2-Z 

Ullah, S., Khan, F. U., & Ahmad, N. (2022). Promoting sustainability through green innovation adoption: a case of manufacturing industry. Environmental Science and Pollution Research, 1-21. 

Von Briel, F., Schneider, C., & Lowry, P. B. (2019). Absorbing knowledge from and with external partners: The role of social integration mechanisms. Decision sciences, 50(1), 7-45. https://doi.org/10.1111/deci.12314

Wang, C., Ren, X., Jiang, X., & Chen, G. (2023). In the context of mass entrepreneurship network embeddedness and entrepreneurial innovation performance of high-tech enterprises in Guangdong province. Management Decision. https://doi.org/10.1108/md-04-2023-0531

Wang, G., & Liu, Z.-B. (2010). What collective? Collectivism and relationalism from a Chinese perspective. Chinese Journal of Communication, 3(1), 42-63. 

Wang, H., Qi, S., Zhou, C., Zhou, J., & Huang, X. (2022). Green credit policy, government behavior and green innovation quality of enterprises. Journal of Cleaner Production, 331, 129834. 

Wang, S., Zhang, M., & Chen, F. (2023). Unveiling the impact of Belt and Road Initiative on green innovation: empirical evidence from Chinese manufacturing enterprises. Environmental Science and Pollution Research, 30(37), 88213-88232. 

Xu, Y., Dong, B., & Chen, Z. (2022). Can foreign trade and technological innovation affect green development: Evidence from countries along the Belt and Road. Economic Change and Restructuring, 55(2), 1063-1090. 

Yuan, S., & Pan, X. (2023). The effects of digital technology application and supply chain management on corporate circular economy: A dynamic capability view. J Environ Manage, 341, 118082. https://doi.org/10.1016/j.jenvman.2023.118082

Zahra, S. A., & George, G. (2002). Absorptive capacity: A review, reconceptualization, and extension. Academy of Management Review, 27(2), 185-203. https://doi.org/10.5465/amr.2002.6587995

Zhang, R., Tai, H., Cheng, K.-T., Cao, Z., Dong, H., & Hou, J. (2022). Analysis on evolution characteristics and dynamic mechanism of urban green innovation network: A case study of Yangtze River economic belt. Sustainability, 14(1), 297. https://doi.org/10.3390/su14010297

---

## [Editor Report · Decision Letter 1]

14 Mar 2024

Does Structure Social Capital Lead to a Proactive Green Innovation? A Three-Part Serial Mediation Model

PONE-D-23-43928R1

Dear Dr. Gao,

We’re pleased to inform you that your manuscript has been judged scientifically suitable for publication and will be formally accepted for publication once it meets all outstanding technical requirements.

Kind regards,

Bablu Kumar Dhar, PhD, Post Doc

Academic Editor

PLOS ONE
---

## [Editor Report · Acceptance letter]

25 Mar 2024

PONE-D-23-43928R1 

PLOS ONE

Dear Dr. Gao, 

I'm pleased to inform you that your manuscript has been deemed suitable for publication in PLOS ONE. Congratulations! Your manuscript is now being handed over to our production team.

Kind regards, 

on behalf of

Dr. Bablu Kumar Dhar 

Academic Editor

PLOS ONE